# Design optimization of high-sensitivity PCF-SPR biosensor using machine learning and explainable AI

**Mst. Rokeya Khatun** *, **Md. Saiful Islam**

Institute of Information and Communication Technology (IICT), Bangladesh University of Engineering and Technology (BUET), Dhaka, Bangladesh

* rokeya2kcse@gmail.com

## Abstract

Photonic crystal fiber based surface plasmon resonance (PCF-SPR) biosensors are sophisticated optical sensing platforms that enable precise detection of minute refractive index (RI) variations for various applications. This study introduces a highly sensitive, low-loss, and simply designed PCF-SPR biosensor for label-free analyte detection, operating across a broad RI range of 1.31 to 1.42. In addition to conventional methods, machine learning (ML) regression techniques were integrated to predict key optical properties, while explainable AI (XAI) methods, particularly Shapley Additive exPlanations (SHAP), were used to analyze model outputs and identify the most influential design parameters. This hybrid approach significantly accelerates sensor optimization, reduces computational costs, and improves design efficiency compared to conventional methods. The proposed biosensor achieves impressive performance metrics, including a maximum wavelength sensitivity of 125,000 nm/RIU, amplitude sensitivity of $-1422.34$ $RIU^{-1}$, resolution of $8 \times 10^{-7}$ RIU, and a figure of merit (FOM) of 2112.15. ML models demonstrated high predictive accuracy for effective index, confinement loss, and amplitude sensitivity. SHAP analysis revealed that wavelength, analyte refractive index, gold thickness, and pitch are the most critical factors influencing sensor performance. The combination of a simple yet efficient design and advanced ML-driven optimization makes this biosensor a promising candidate for high-precision medical diagnostics, particularly cancer cell detection, and chemical sensing applications.

## 1. Introduction

Photonic crystal fiber-based surface plasmon resonance (PCF-SPR) biosensors combine both photonic crystal fiber (PCF) surface plasmon resonance (SPR) techniques to provide very sensitive and ideal optical characteristics. Knight et al. conceptualized PCF in 1996 [1], marking a significant breakthrough in optical fiber

**Data availability statement:** All relevant data for this study are publicly available from the GitHub repository (https://github.com/MRKhatun/PCF_SPR_Biosensor_Design_Optimization).

**Funding:** The author(s) received no specific funding for this work.

**Competing interests:** The authors have declared that no competing interests exist.

technology. Its unique design is characterized by both the core and the cladding areas, which have a regular pattern of air holes along the length of the core [2]. Comparatively, PCF outshines conventional fibers in terms of their optical properties, such as chromatic dispersion, birefringence, and confinement loss (CL) [3,4]. SPR is an optical phenomenon that occurs when incident light excites collective electron oscillations at the interface of a metal and a dielectric medium, enabling highly sensitive detection of refractive index (RI) changes. The material composition and design variation of PCF-SPR biosensors greatly influence their sensitivity performances. Various studies explored diverse sensing mechanisms using PCF-SPR biosensor technology [5–7]. To get the best performance, the parameters like the air hole radius in core cladding areas, pitch distance (p), gold layer thickness ($t_g$), perfectly matched layer (PML) thickness, and analyte layer thickness are optimized. Gold and silver are the most-used materials for plasmonic applications due to their high electron densities. Silver is better for plasmonic electronics because it is a better conductor, while gold is better for strong plasmonic resonance applications because it is more stable chemically and has a higher absorption coefficient [8,9]. PCF-SPR sensors are renowned for their exceptional sensitivity, compact dimensions, and versatility. To optimize the performance of PCF-SPR biosensors, it is essential to minimize CL and enhance sensitivity. The enhancement of sensor performance remains a focus for exceptional efficacy, and finding them applicable in diverse scientific and industrial domains [10,11]. Specifically, PCF-SPR sensors are widely applied in diverse fields such as medical diagnostics, biochemical sensing, detection of bioorganic compounds, and various clinical testing procedures [12,13].

PCF-SPR biosensor development has advanced significantly in recent years to explore high sensitivity characteristics. Recently, researchers have been using machine learning (ML) approaches to evaluate SPR sensor performance based on different input features, especially design parameters. Researchers have focused on addressing the complexity and minimizing the time-intensive nature of simulations to explore high-performance SPR-PCF biosensor models [14–16]. Traditional ML models are being used to optimize the performance of PCF-SPR performance. Also, artificial neural networks (ANNs) help accurately guess the optical features of PCF, such as its effective index, mode area, dispersion, and CL [17]. ML training and testing are faster than traditional methods like numerical MODE solutions [18]. This makes it possible to accurately predict outcomes for various PCF parameters. Furthermore, the research [19] shows a fast deep learning model for predicting optical properties in photonic sensors, with low mean square error (MSE) outperforming conventional methods. Another study presents a COMSOL Multiphysics simulation along with a novel artificial intelligence (AI)-driven method for optimizing PCF design [20]. Researchers have investigated the application of support vector machines (SVMs) as a substitute for ANNs in forecasting the optical properties of PCF. For this case, they explored optical biosensors with a design error rate of less than 3% shows that ML-based methods have been used to improve their performance. ML applications for structural optimization have been emerging to improve PCF-SPR biosensors in order to make them more sensitive and CL. In order to enhance

performance, a study by [21] investigated a PCF-SPR sensor that used ANN. Where the sensor achieved an wavelength sensitivity ($S_\lambda$) of 18,000 nm/RIU, an amplitude sensitivity ($S_A$) of 889.89 RIU$^{-1}$, and a resolution of $5.56 \times 10^{-6}$ RIU for analyte RI ($n_a$) 1.31 to 1.4. Similarly, [22] developed a high-sensitivity PCF-SPR biosensor for cancer detection, achieving a figure of merit (FOM) of 36.52 RIU$^{-1}$ and an $S_\lambda$ of 13,257.20 nm/RIU. All of this research demonstrates how well ML works to optimize PCF-SPR biosensors, which makes them ideal for chemical, pharmacological, and biological sensing applications.

Although PCF-SPR biosensors exhibit considerable potential in sensing applications, enhancing their performance has a significant barrier [23]. The main challenges involve ensuring an optimal balance among sensitivity, accuracy, and signal loss and providing compatibility with various analyte. Several studies have systematically investigated methods to improve the efficiency and performance of PCF-SPR biosensors [24–26]. Recent studies show that combining ML with PCF-SPR biosensor technology is a useful way to improve sensor performance. Researchers have shown that ML methods can predict the operating parameters and optimal design in an efficient way. This takes a lot less time than traditional simulation techniques. Despite that, the utilization of ML in PCF-SPR biosensor design remains emerging. Only a few researches have been done that use ML-based modelling for high-performance biosensor design optimization. To the best of our knowledge, no previous study has integrated explainable artificial intelligence (XAI) with ML for the optimization of PCF-SPR biosensor designs. This gap in the literature review highlights the need for an extensive study that will focus on the efficient, high-performance PCF-SPR biosensor design. This study aims to fill up this gap by carefully looking at the important design factors and performance of PCF-SPR biosensors. We focused on performance improvement in key areas like sensitivity, resolution, and FOM with a simple designed sensor. This study not only looks at how to improve sensors, but it also shows how input parameters can change the properties of sensors using XAI. The primary objectives of our research work are:

- To explore a highly sensitive, efficient PCF-SPR biosensor design characterized by low CL, high $S_A$, $S_\lambda$, Resolution and FOM.

- To employ diverse ML methods to enhance the predictive power of biosensor properties by optimizing accuracy and reducing error.

- To employ XAI techniques, such as SHAP, to uncover the influence of design parameters, enabling a more transparent and interpretable biosensor design process.

Here's how we present our research: Section 2 outlines the study's materials, methods, design parameters, simulation process, and the ML techniques. Section 3 presents result and analysis of our study, comparing them with key findings. Section 4 provides a discussion, while Section 5 concludes the study by summarizing our findings and suggesting directions for future research.

## 2. Materials and methods

This investigation utilizes a structured approach that combines biosensor design, optical simulations, ML, and XAI to explore a PCF-SPR biosensor. The research workflow for PCF-SPR biosensor model optimization, using ML applications, is illustrated in Fig 1. The initial phase involves the design of the PCF-SPR biosensor. Then we performed sensor design and simulations using COMSOL Multiphysics to evaluate essential properties, including effective refractive index ($N_{eff}$), CL, $S_A$, $S_\lambda$, Resolution and FOM. Next, we generated and preserved the data from these simulations for future analysis. Subsequently, we utilize ML algorithms to predict and improve the performance of the sensor. Multiple regression models have been employed, including random forest regression (RF), decision tree (DT), gradient boosting (GB), extreme gradient boosting (XGB), and bagging regressor (BR). These models aid in uncovering patterns and correlations among various design parameters and the attributes that are being optimized. This speeds up the optimization process and improves its

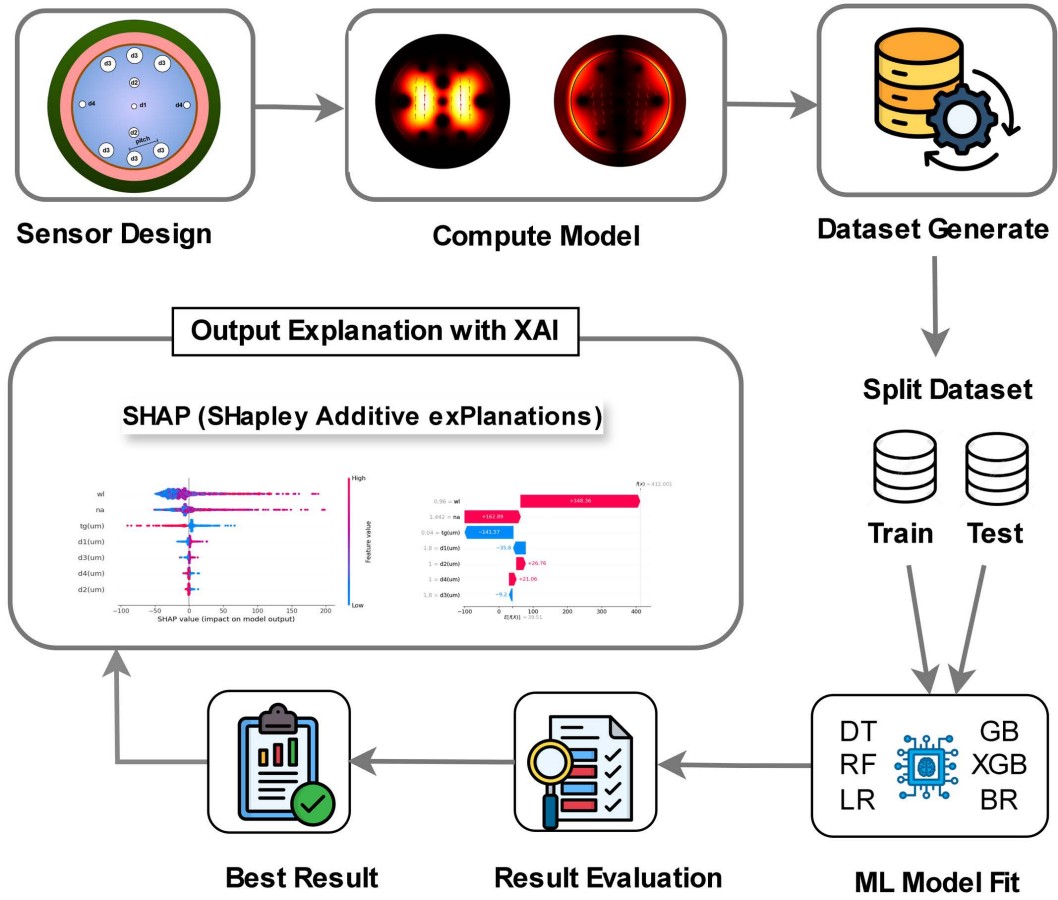

**Fig 1. Research workflow for PCF-SPR biosensor model optimization.**

effectiveness relative to traditional methods. XAI methodologies like SHAP are utilized to examine how different parameters influence the performance of the sensor. This allows for the modification and enhancement of the design based on insights derived from data. In conclusion, we evaluate the accuracy of the ML models through metrics such as R-squared ($R^2$), mean absolute error (MAE), and MSE.

## 2.1 Working principle of PCF-SPR biosensor

Fig 2 illustrates the working principle of an optical fiber-based sensing system that relies on evanescent field interaction for detection. A broadband light source injects light into the core of a specially designed optical fiber, where it propagates through total internal reflection. At each reflection point, a portion of the light extends beyond the core into the cladding region as an evanescent wave. A syringe injection pump introduces the target sample (such as a chemical or biomolecule solution) into a flow chamber that surrounds the sensitive region of the fiber. When the evanescent field interacts with the sample, any change in the local RI, absorption, or scattering alters the transmitted light's properties. This modified optical signal is collected at the output end and sent to an optical spectrum analyzer (OSA), which measures variations in the transmission spectrum (e.g., wavelength shift or intensity change). The data is then processed and analyzed using computer software to detect and quantify the presence of the analyte, making the system highly suitable for label-free and real-time sensing applications.

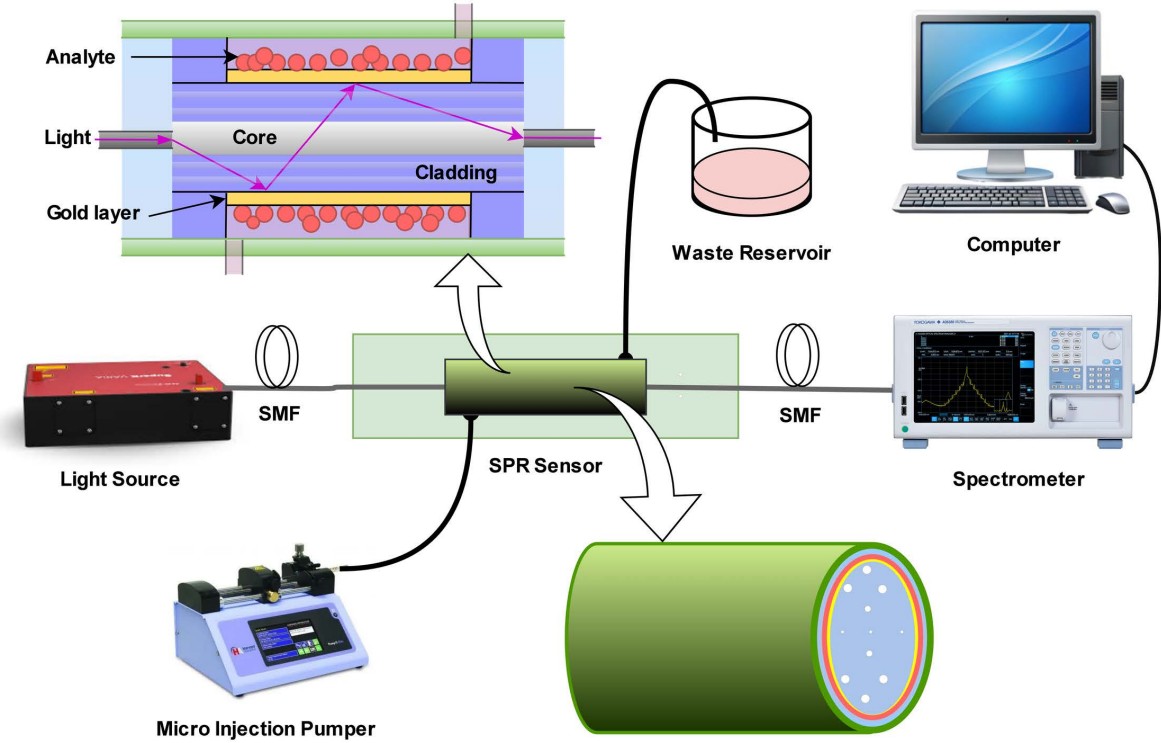

**Fig 2. Schematic diagram of a Fiber-Optic SPR Biosensor system for biochemical detection.**

The proposed PCF-SPR biosensor is engineered for effective real-time sensing in both medical diagnostics and environmental monitoring. In the medical domain, it is capable of detecting a wide range of biomarkers, including cancer indicators, hormones, proteins, and pathogens such as viruses and bacteria in aqueous biofluids like blood and serum. Its high sensitivity and broad $n_a$ detection range (1.31–1.42) make it particularly suitable for early-stage disease diagnosis and continuous health monitoring. This RI range covers most biological analytes, typically found in environments with $n_a$ values between 1.33 and 1.40, while the inclusion of the 1.31 to 1.32 range, though less common in practical biosensing, allows for theoretical analysis of sensor behavior near the lower detection threshold. This supports a more comprehensive understanding of sensitivity performance and helps evaluate the design's robustness under edge-case or simulation-driven conditions. In environmental applications, the sensor can be employed to identify chemical contaminants, solvents, and pollutants in water or air samples. Its fast response time, compatibility with microfluidic systems, and adaptability to portable and point-of-care platforms make it highly suitable for on-site detection, addressing critical challenges in both clinical and environmental settings.

## 2.2 Sensor cross-sectional design

The proposed biosensor optimization approach involves simulation with defined geometric features, incorporating components like gold layers, a fused silica substrate, air cavities, and an analyte zone. Fig 3 presents the cross-sectional structure of a PCF based SPR biosensor, developed for precise biochemical detection. The design with a smaller number of air holes reduces fabrication complexity. In our design, we have utilized only 11 air holes, a significantly smaller number compared to previous redesigns. The fiber consists of a core surrounded by a structured cladding, where periodically arranged air holes (d1, d2) help guide light through the fiber. The spacing between these air holes is defined by the pitch (p), ensuring proper light propagation. Key structural distances include D1 = 2.6 × p, D2 = 3 × p, and D3 = 3.5 × p, which

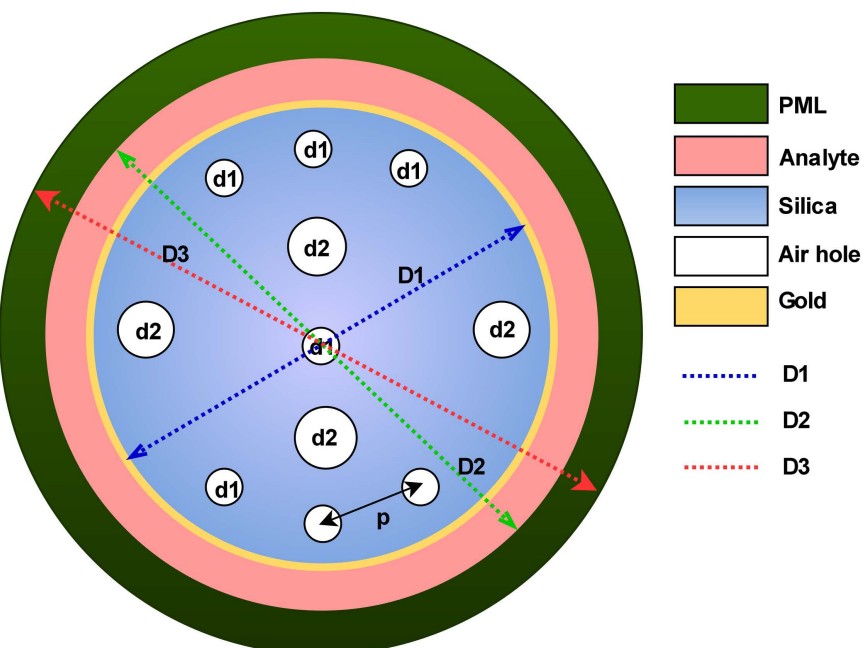

**Fig 3. Cross-sectional diagram of a PCF-Based SPR biosensor.**

represent the radial placement of different regions in the fiber structure. D1 extends from the fiber's center to the gold layer, the region where SPR takes place. D2 reaches the analyte layer, where interactions between biomolecules and the sensing surface occur. D3 extends further to the PML, which absorbs unwanted radiation and minimizes reflection. These structural parameters are essential for optimizing light transmission, improving sensor performance, and enhancing detection sensitivity. The color-coded regions represent different fiber components.

## 2.3 Methodologies

The performance of a PCF-SPR biosensor depends on several key parameters [27]. $N_{eff}$ ensures proper light propagation and phase matching. While CL measures optical power leakage, lower values indicate better efficiency. $S_A$ enhances detection by quantifying intensity changes, and $S_\lambda$ detects small RI variations by measuring resonance wavelength shifts. Resolution determines the smallest detectable RI change, with higher resolutions offering more precision. The FOM balances sensitivity and resonance sharpness, ensuring accurate detection. These parameters collectively evaluate sensor performance for biomedical, chemical, and environmental applications. Moreover, the surface plasmon polariton (SPP) in PCF SPR biosensors states electromagnetic waves that move over the metal-dielectric interface and are produced when light combines with free electrons on the metal surface. The phenomenon that generates SPP is called SPR. These waves are highly sensitive to changes in the $n_a$, which makes them crucial in PCF-SPR biosensors for analyte detection. Conversely, the fundamental mode directs light through the PCF, facilitating its propagation within the sensor. Here we are giving the brief overview of different optical parameters for the PCF-SPR biosensor.

The Sellmeier equation defines the calculation of the RI of fused silica, based upon the wavelength of light [28]. The equation is stated as follows:

$$n^2(\lambda) = 1 + \frac{B_1 \lambda^2}{\lambda^2 - C_1} + \frac{B_2 \lambda^2}{\lambda^2 - C_2} + \frac{B_3 \lambda^2}{\lambda^2 - C_3}$$

(1)

The Sellmeier model utilizes specific constants to characterize the wavelength-dependent RI of materials. The constants used in this context are: B1 = 0.69616300, B2 = 0.407942600, B3 = 0.897479400, and C1 = 0.00467914826, C2 = 0.0135120631, C3 = 97.9340025.

The CL is denoted by the symbol $\alpha$(dB/cm) [28]. CL in the sensor is computed according to the following relation, using the imaginary part of the $N_{eff}$:

$$\alpha\left(\frac{dB}{cm}\right) = 8.686 \times k_0 Im(N_{eff}) \times 10^4 \tag{2}$$

Where $k_0 = 2\pi/\lambda$ represents the free-space wave number, and $Im(N_{eff})$ is the imaginary component of the $N_{eff}$. This expression quantifies optical attenuation resulting from imperfect mode confinement, which plays a crucial role in sensor characterization.

Amplitude sensitivity ($S_A$) is another vital property that indicates the sensor's response with the variations of $n_a$ by observing changes in transmitted light intensity [29]. The numerical formulation is expressed as:

$$S_A = \frac{1}{\alpha(\lambda, n_a)} \cdot \frac{\partial\alpha(\lambda, n_a)}{\partial n_a} \tag{3}$$

Here, $n_a$ indicates the RI of the analyte, and $\partial n_a$ represents the difference between the $n_a$ of two neighboring analytes. The function $\alpha(\lambda, n_a)$ denotes the CL, while $\partial\alpha(\lambda, n_a)$ presents the variation in CL as the $n_a$ changes. This formula evaluates the effect of $n_a$ fluctuations on the amplitude of light waves through the SPR sensor.

Another critical performance metric is wavelength sensitivity ($S_\lambda$), which reflects how the sensor output responds to changes in the incident light's wavelength [29]. It is defined by the following relation:

$$S_\lambda = \frac{\Delta\lambda_{peak}}{\Delta n_a} \tag{4}$$

Where $\Delta\lambda_{peak}$ represents the peak wavelength variation, and $\Delta n_a$ is the variation in $n_a$ between two adjacent analytes.

The calculation of sensor resolution is essential to assess the detection capability of the proposed sensor, as defined by the equation below [30]:

$$R\ (RIU) = \Delta n_a \times \frac{\Delta\lambda_{min}}{\Delta\lambda_{peak}} \tag{5}$$

Where $\Delta n_a$ represents the change in $n_a$, and $\Delta\lambda_{min} = 0.1$ nm denotes the smallest detectable wavelength shift. This equation helps quantify the sensor's ability to distinguish small variations in analyte concentration.

The Figure of Merit (FOM) is an important parameter used to evaluate the performance of an SPR-based biosensor. It is defined as the ratio of $S_\lambda$ to the full width at half maximum (FWHM) of the resonance curve [30]. The formula for FOM is given by:

$$FOM = \frac{S_\lambda}{FWHM} \tag{6}$$

Where, $S_\lambda$ (nm/RIU) indicates the shift in resonance wavelength per unit change in $n_a$, and FWHM (nm) representing the spectral width of the resonance dip.

For a high-performance PCF-SPR biosensor, higher values of FOM, $S_\lambda$, and $S_A$ contribute to improved detection accuracy, enhanced signal response, and better analyte sensitivity. In contrast, lower CL improves light propagation, minimizes

signal attenuation, and enhances overall sensor efficiency. The mapping of optical parameters and their impact on biosensor performance is given below.

- Higher FOM → Sharper resonance peak → Improved detection accuracy and resolution

- Higher $S_A$ → Greater intensity variations → More precise signal detection

- Higher $S_\lambda$ → Larger resonance wavelength shift → Enhanced sensitivity to analyte changes

- Lower CL → Better light propagation → Reduced signal attenuation and improved efficiency

## 2.4 Dataset generation

In this study, we have used COMSOL Multiphysics software to design and simulate the sensor models and evaluated these models using the finite element method (FEM) [31]. The dataset is derived from simulation outputs and includes critical structural and optical features such as core diameters (d1, d2), spacing between air holes pitch (p), thickness of the gold coating ($t_g$), $n_a$, operational wavelength (wl), calculated $N_{eff}$, CL, and $S_A$. In addition to these primary features, we derived further optical performance metrics such as $S_\lambda$, resolution, and FOM. This comprehensive dataset was also utilized to train ML models aimed at predicting key optical responses of the proposed sensor across various structural configurations, thereby facilitating the identification of high-performance design combinations.

To evaluate each single-mode calculation simulation took around 2 minutes. The amount of time varies according to the number of degrees of freedom resolved. The number of degrees of freedom solved for our mode analysis is 242881. This number depends on the structural design of the sensor, particularly the size and arrangement of the air holes. We used a workstation with a 2.3 GHz Intel Core i3 processor and 12 GB of RAM to perform the calculations.

The simulation process generally follows these steps:

- Build the biosensor design according to the plan.

- Choose suitable electromagnetic interfaces for the analysis.

- Define the materials used for the core and cladding.

- Create a mesh to ensure precise estimates.

- Choose the input settings.

- Use a suitable solver to run the exercise.

- Evaluate the results and explain what the data means.

- Change the settings if necessary and improve the model.

- Gather the results and create a dataset for more research.

Here, Fig 4 represents different states of the PCF-SPR biosensor model, depending on its design and field distributions. Here, Fig 4(a) shows the meshed model of our biosensor model. That divides the computational domain into small, finite elements to solve partial differential equations (PDEs) for light propagation using the FEM [32]. The accuracy and efficiency of the simulation depend on the quality of the mesh. That depends on the air hole arrangement and other design parameters of the sensor structure. Again, Fig 4(b) illustrates the electric field distribution of the core mode for the y-polarized mode. The bright regions indicate areas of strong electric field intensity, revealing how light interacts within the fiber. Fig 4(c) displays the power flow distribution of the SPP mode for the y-polarized mode. The energy is mostly gathered near the plasmonic interface. This shows that the core mode and the plasmonic mode are coupled, which is important for sensing purposes.

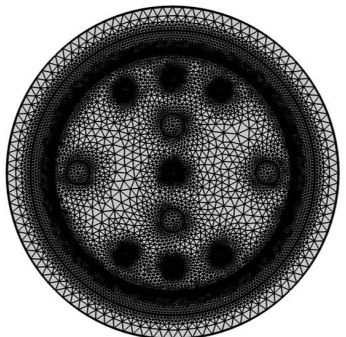 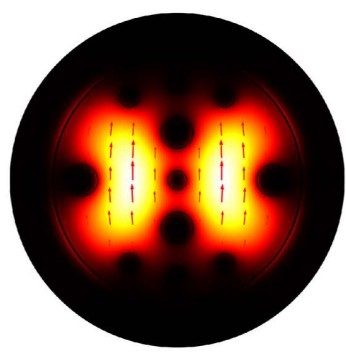 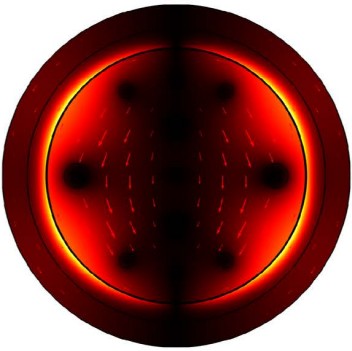

**Fig 4. PCF-SPR biosensor: (a) meshed model, (b) core mode electric field distribution, and (c) SPP mode power flow distribution for y-polarized mode.**

This dataset employs parameters such as d1, d2, p, $t_g$, $n_a$, and wl as predictor variables. The outputs or target attributes include the $N_{eff}$, CL, $S_A$, and $S_\lambda$. From the simulation result, we have got $N_{eff}$ and then numerically calculated the optical properties. The data provides insights into how structural modifications influence light propagation and resonance behavior of the sensor. This dataset provides an important insight for PCF-SPR biosensor performance optimization. Table 1 presents an overview of the main variables included in the dataset used for analyzing the PCF-SPR biosensor.

## 2.5 AI techniques for PCF-SPR sensor optimization

Prior studies [33] mentioned that SPR-based sensors are well-suited for label-free detection of cancer markers due to their ability to monitor RI variations resulting from specific molecular binding events. AI techniques, particularly ML, have emerged as a powerful tool in the design and optimization of PCF-SPR biosensors. These data-driven models facilitate rapid prediction of complex optical properties, such as $N_{eff}$, CL, and $S_A$, based on sensor design parameters. That significantly reduces the time-consuming simulation processes. ML models, including RF, SVM, GB, and Bayesian regularization neural networks (BRANNs), have shown potential in modeling nonlinear relationships between structural features and sensor performance. Such approaches enable efficient inverse design, structural tuning, and enhanced prediction fidelity, making them ideal for biosensing applications. Furthermore, integrating ML with XAI enhances both optimization speed and interpretability. Based on other studies [34], different AI techniques can be used for sensor performance optimization as follows:

**Table 1. Description of dataset variables used in PCF-SPR biosensor modeling.**

| Feature | Explanation | Example Value/ Unit |
|---|---|---|
| d1 | Small air hole diameter (μm) | 0.6, 1.2, 1.4, 1.6 |
| d2 | Large air hole diameter(μm) | 0.6, 0.8, 1 |
| p | Pitch (distance between two air holes) (μm) | 2, 2.5, 3 |
| $t_g$ | Thickness of gold (nm) | 30, 40, 50, 60 |
| $n_a$ | The RI of the analyte | 1.31-1.42 |
| wl | Wavelength (μm) | 0.4–1.35 |
| $N_{eff}$ | Core mode $N_{eff}$ on y-axis | 1.4284–1.47 |
| CL | Confinement loss (dB/cm) | 1.614–437.15 |
| $S_A$ | Amplitude sensitivity (RIU$^{-1}$) | −1422.34–95.78 |

BRANNs are capable of learning from small, noisy datasets and capturing complex nonlinear dependencies between input parameters and output targets. The method reduces overfitting and enhances generalization through regularization, making it ideal for high-accuracy, real-time applications and inverse sensor design. Classical ML models such as SVM, DT, and ensemble methods like RF, AdaBoost (AB), and GB have been used for classifying optical modes and predicting sensor behavior. These models are particularly useful in early-stage design to efficiently evaluate large design spaces and estimate key performance indicators. Furthermore, advanced deep learning architectures, including feedforward neural networks (FNNs), convolutional neural networks (CNNs), and generative models such as variational autoencoders (VAEs) and conditional generative adversarial networks (GAN), are now being applied for precise resonance prediction, analyte classification, and automated inverse design of PCF structures. CNNs can analyze mode profile images, while autoencoders and 1D-CNNs help in denoising experimental data. Deep reinforcement learning (DRL) offers an adaptive framework for multi-objective optimization under fabrication and material constraints. It enables the sensor to learn from operational history and adjust design parameters in real-time, thereby supporting self-calibrating and self-optimizing systems. Adaptive sensing and environmental compensation AI techniques can also be employed to monitor environmental drift and sensor degradation. By learning from prior data, these models can dynamically adjust sensing parameters to maintain consistent accuracy over extended periods, which is especially critical in remote or in vivo diagnostics. So it is clear that the integration of these AI techniques not only enhances predictive accuracy but also reduces design time and computational cost, thus enabling the rapid development of next-generation PCF-SPR biosensors with enhanced sensitivity, robustness, and adaptability.

## 2.6 ML algorithms

Since our target variables are continuous, regression models were the appropriate choice for prediction tasks. We applied various regression algorithms to analyze the dataset and assess how design and other factors impact the outcomes. Below is an overview of the ML regression techniques employed:

Decision Tree Regression (DTR) operates by dividing the input feature space into distinct regions and making predictions based on the average target value within each region [35]. This supervised, non-parametric approach is intuitive and allows straightforward visualization of decision rules. However, if left unchecked, it can easily overfit the training data.

Random Forest Regression (RFR) builds an ensemble of decision trees by training each tree on a different random subset of the data and selecting random features at each split [36,37]. The final prediction is obtained by averaging the outputs of all individual trees. This approach enhances prediction stability and accuracy while effectively reducing the risk of overfitting, making it well-suited for handling complex and varied datasets.

Gradient Boosting Regression (GBR) sequentially constructs an ensemble of weak learners, where each subsequent model aims to correct the residual errors of its predecessors [38]. This iterative refinement enables GBR to model complex patterns effectively and achieve high prediction accuracy.

Extreme Gradient Boosting Regressor (XGBR) extends GB by optimizing both computational speed and predictive performance [39]. It combines a series of decision trees trained to minimize errors, excelling at handling large-scale data and intricate relationships within features.

BR improves model stability by training several base regressors on different randomly sampled subsets of data with replacement [40]. This ensemble method reduces variance and enhances prediction accuracy. The ensemble's output is determined by combining the predictions of all base models through averaging, which helps minimize prediction variability and mitigates the risk of overfitting.

The integration of ML techniques plays a crucial role in predicting the optical properties of the proposed PCF-SPR biosensor, such as $N_{eff}$, CL, and $S_A$. These ML models effectively learn the nonlinear mapping between key design parameters (e.g., $n_a$, $t_g$, pitch, air-hole diameters) and corresponding optical properties. This not only reduces computational time from minutes to milliseconds but also allows for rapid screening of multiple design combinations. Among the tested

models, RFR and XGBR provided the excellent trade-off between prediction accuracy and execution time. Furthermore, ML approaches aid in precise finalization of sensor parameters based on maximum sensitivity, minimal CL, and high FOM.

## 2.7 ML models evaluation metrics

ML regression model performance evaluation needs to assess using different performance metrics [41]. Here the metrics description will be described in short to understand the implication and numerical calculation process. Smaller MSE values indicate a more accurate model, as they reflect smaller differences between predicted and actual outcomes. MSE evaluates how well a model predicts by quantifying the average squared gap between actual outcomes and predicted values. It reflects how much the predictions deviate from the true values, with lower MSE indicating better model accuracy. The metric is calculated by averaging the squares of all prediction errors across the dataset [41].

$$MSE = \frac{1}{N} \sum_{i=1}^{N} (y_i - \hat{y})^2$$

(7)

Here, $\hat{y}$ stands for the model's forecasted value, while y denotes the actual observed value.

A MAE reflects higher predictive accuracy, as it captures the average magnitude of deviation between forecasts and actual outcomes. MAE is computed by averaging the absolute errors across all predictions, providing a straightforward indicator of prediction quality [41]. The equation (8) presents the formula for MAE calculation.

$$MAE = \frac{1}{N} \sum_{i=1}^{N} |y_i - \hat{y}_i|$$

(8)

In this context, N denotes the total number of data points and $y_i$ represents the observed value, while $\hat{y}_i$ represents the predicted value.

The $R^2$ value quantifies the efficacy of a model in predicting the dependent variable's outcome. The range is from 0 to 1, with 0 indicating no explanation of the variance in the dependent variable and 1 indicating a complete explanation of the relationships. An $R^2$ value of 1 indicates that the model accounts for 100% of the variance in the target variable, while a score of 0.5 suggests that the model explains 50% of the variance [41].

We use the following formula to compute the $R^2$ value:

$$R^2 = 1 - \frac{\sum_{i=1}^{n} (y_i - \hat{y}_i)^2}{\sum_{i=1}^{n} (y_i - y)^2}$$

(9)

In this expression, the numerator $\sum(y_i - \hat{y}_i)^2$ represents the total squared error between the predicted values $\hat{y}_i$ and the actual observations $y_i$. The denominator $\sum(y_i - \bar{y})^2$ captures the total variation in the actual data around the mean value $\bar{y}$.

## 2.8 Explainable AI (XAI)

XAI that makes AI models easier to understand and interpret. It focuses on developing methods that make the decisions and processes of AI systems transparent, so users can make sense of how outcomes are reached. This is crucial in areas like healthcare, finance, and automotive industries, where the accuracy and reliability of AI predictions are critical [42–44]. SHAP is one of the most widely used methods for model interpretability. It provides both global explanations that capture the overall behavior of the model and local explanations that detail how individual features influence specific predictions. SHAP assigns each feature a value representing its contribution to the prediction. These values are consistent and additive, ensuring that the sum of all feature contributions equals the difference between the predicted value and the model's average output, thereby offering a fair and transparent measure of feature importance [45].

To enhance the transparency and trustworthiness of the ML model used for predicting optical properties, SHAP was employed as an XAI technique. SHAP provides detailed insights into how each input parameter, such as air hole diameters, pitch, $t_g$, and $n_a$ contributes to model outputs like $N_{eff}$, CL, and $S_A$. By visualizing SHAP values, we were able to identify the most influential design features, enabling more informed and efficient adjustments to the PCF-SPR biosensor structure. While SHAP does not directly enhance the sensor's physical sensing performance, it plays a crucial role in interpreting the ML model's predictions. This interpretability supports targeted design optimization, reduces the dependency on exhaustive simulations, and fosters the decision-making process for sensor performance optimization.

## 3. Results & analysis

### 3.1 Analysis of simulation results

Fig 5 displays the coupling point where the surface SPP mode intersects with the core mode. The structure features air hole diameters of 0.6 µm for d1 and 1 µm for d2, with a $t_g$ of 40 nm. We have used the COMSOL Multiphysics simulation to evaluate $N_{eff}$ real and imaginary values along the y-axis that were used to calculate various propagation properties. Fig 5 displays the coupling point where the surface SPP mode intersects with the core mode under the conditions: $n_a$ of 1.37, core diameters d1 = 0.6 µm and d2 = 1 µm, and a $t_g$ = 40 nm. The figure demonstrates how loss increases with the decrease in wavelengths and then decreases as they increase. The figure also shows how $N_{eff}$ varies with wavelength, indicating a decrease in $N_{eff}$ as the wavelength increases. At a wavelength of 0.67 µm, the real parts of $N_{eff}$ for the y-polarized core and SPP modes coincide, marking the phase matching condition. This point is significant as SPR occurs here. At this wavelength, CL reaches its maximum, and $N_{eff}$ is 1.451. The graphs in Fig 6 illustrate how changes in d1 and d2 size influence CL variation with wavelength, where p = 2 µm, $t_g$ = 40 nm, and $n_a$ = 1.37. In Fig 6(a), where d2 is fixed at 1 µm, d1 increases gradually from 0.6 µm to 1.6 µm. With the increase of d1, CL also increases gradually. The lowest loss is observed when d1 is 0.6 µm, while larger values of d1 result in higher loss. Similarly, graph in Fig 6(b), where d1 is fixed at 1.2 µm and d2 changes from 0.6 µm to 1.2 µm. In this case, the highest CL is seen at d2 = 0.6 µm, and the loss goes down as d2 increases.

Furthermore, Fig 7 illustrates the relationship between $S_A$ and wavelength for different air hole sizes. We kept the other parameters fixed at p = 2µm, $t_g$ = 40 nm, and $n_a$ = 1.37. From the graphs, it is clear that the variation in air hole dimensions influences the sensor's response, which causes changes in $S_A$. In Fig 7(a), $S_A$ is plotted for different d1 values, keeping

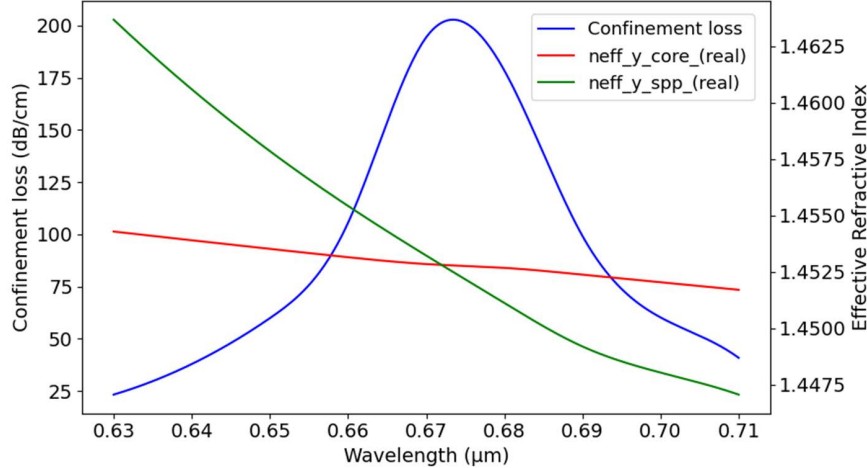

**Fig 5. $N_{eff}$ core and SPP mode phase matching at peak CL.**

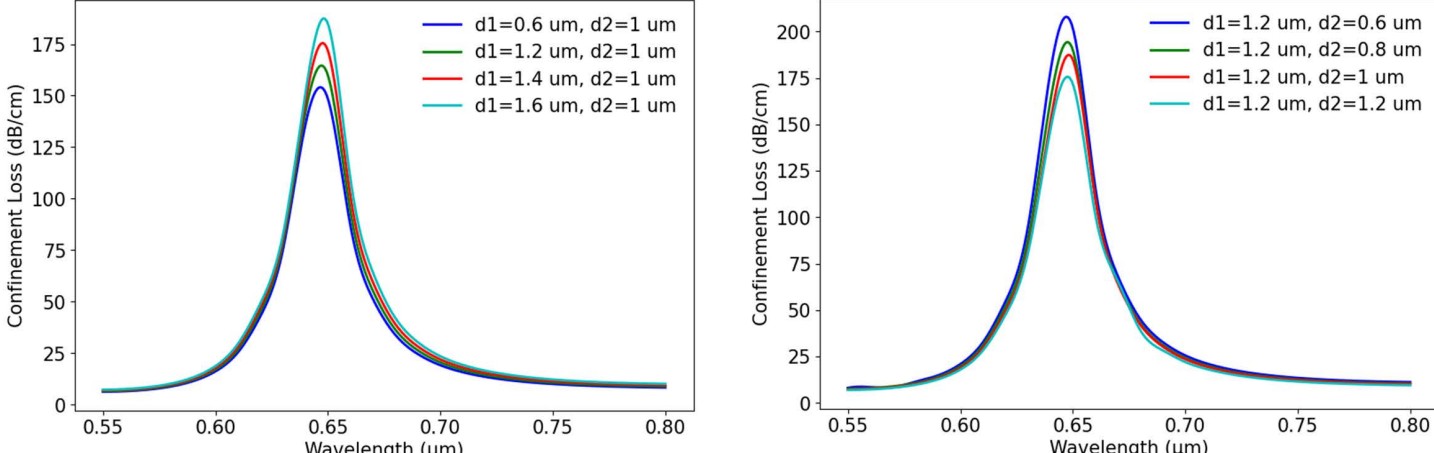

**Fig 6. Effect of d1 and d2 variation on CL (a) wavelength vs. CL with different d1 for fixed d2 = 1 μm (b) wavelength vs. CL with different d2 for fixed d1 = 1.2 μm.**

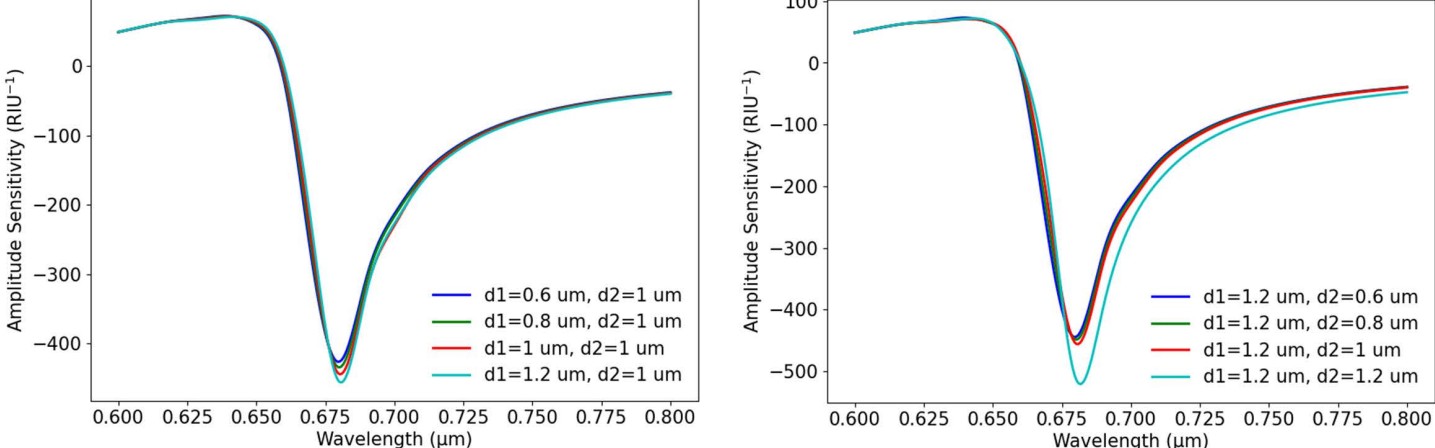

**Fig 7. Effect of air hole diameter variation on $S_A$ (a) $S_A$ vs. wavelength with the changes of d1, (b) $S_A$ vs. wavelength with the changes of d2.**

d2 = 1 μm as a fixed value. From the graph, we get that as d1 increases from 0.6 μm to 1.2 μm, the $S_A$ curve shifts to a higher value. The highest $S_A$ is achieved at d1 = 1.2 μm, indicating that a larger air hole in this design enhances plasmonic interaction and improves sensitivity. Fig 7(b) presents the effect of d2 variation, keeping d1 = 1.2 μm as constant. The graph shows that as d2 increases from 0.6 μm to 1.2 μm, $S_A$ also increases. This suggests that an increase in d2 makes the core-plasmon coupling stronger. Overall, the analysis demonstrates that optimizing d1 and d2 plays a crucial role in achieving high CL and $S_A$, which impacts the sensor's detection capabilities.

Additionally, from the experiment, we depict that the thickness of $t_g$ has a significant effect on sensor performance. We varied $t_g$ from 30nm to 60nm while keeping d1 = 0.6μm, d2 = 1μm, and p = 2μm constant. Fig 8(a) shows how CL changes with wavelength for different values of $t_g$. The graph in Fig 8(a) indicates that as the $t_g$ increases, the CL decreases. For $t_g$ = 30nm (blue curve), the highest CL is observed, with a peak value of 350 dB/cm 0.65μm. When the thickness of $t_g$ increases to 40nm (green curve), the peak CL shifts slightly toward higher wavelengths and reduces intensity. For

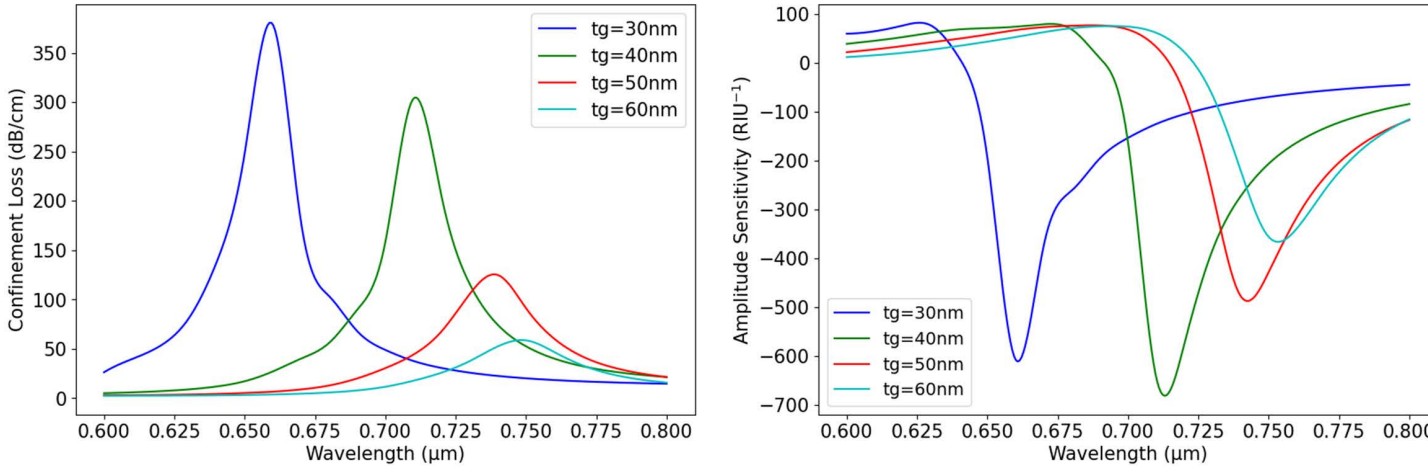

**Fig 8. (a) Wavelength vs. CL with variation of $t_g$ (b) wavelength vs. $S_A$ with different $t_g$.**

$t_g$ = 50nm (red curve), the CL continues to decline, and the peak becomes broader. At $t_g$ = 60nm (cyan curve), the loss is at its lowest value, with a less pronounced peak, indicating a weaker plasmonic effect. This pattern suggests that a thinner gold layer enhances plasmonic resonance, leading to higher CL, while a thicker layer reduces it.

Again, Fig 8(b) presents the impact of different $t_g$ on $S_A$ across varying wavelengths. The graph indicates that as $t_g$ increases, $S_A$ tends to decrease. When $t_g$ = 30 nm (blue curve), the most significant negative $S_A$ is observed 0.66 μm. Increasing the thickness to 40 nm (green curve) results in a slight reduction in sensitivity and a shift in the peak toward a longer wavelength. With $t_g$ = 50 nm (red curve), the sensitivity declines further, and the peak becomes broader. At $t_g$ = 60 nm (cyan curve), the lowest sensitivity is recorded, with a less prominent peak, indicating weaker plasmonic interaction. These findings suggest that a thicker gold layer minimizes CL and improves sensitivity, while a thinner layer reduces the sensor's overall performance.

Pitch size also plays a crucial role in determining the performance of the sensor. In this study, the pitch was varied from 2 μm to 3 μm while keeping $n_a$ = 1.37, d1 = 0.6 μm, d2 = 1 μm, and $t_g$ = 40 nm constant. Fig 9(a) illustrates how CL changes with wavelength for different pitch values. The graph in Fig 9(a) shows that as the pitch increases, CL decreases. At pitch, p = 2 μm (blue curve), the highest CL is observed at wavelength 0.66 μm with high intensity. When the pitch increases to 2.5 μm (green curve), the peak shifts slightly toward a longer wavelength, and the loss intensity reduces. With a pitch of 3 μm (red curve), CL continues to decline, and the peak becomes broader, indicating a weaker plasmonic effect. This suggests that a smaller pitch leads to stronger light confinement and higher loss, while a larger pitch allows better mode propagation and reduces CL. Again, Fig 9(b) examines the effect of pitch variation on $S_A$. A similar trend is observed, where a smaller pitch results in higher sensitivity, with a pitch of 2 μm showing the strongest negative $S_A$ at wavelength 0.66 μm. As the pitch increases to 2.5 μm, the sensitivity decreases slightly, shifting toward a longer wavelength. When the pitch reaches 3 μm, the $S_A$ further weakens, leading to a broader response. These findings indicate that pitch size must be optimized based on the sensor's intended application. We should carefully adjust the pitch choice to balance sensitivity and stability in sensor design.

Fig 10 illustrates the variation in CL across wl for different analyte refractive indices, keeping d1 = 0.6 μm, d2 = 1 μm, p = 3 μm, and gold thickness at 60 nm constant. This analysis provides insights into how $n_a$ influences the optical behaviour of the sensor. In Fig 10(a), CL rises with wavelength, displaying distinct peaks for $n_a$ values between 1.31 and 1.41. As $n_a$ increases, the resonance peak shifts to longer wavelengths, and the peak intensity gradually increasing. This shift reflects stronger plasmonic coupling at higher $n_a$ values, resulting in greater CL. The trend underscores the strong dependence

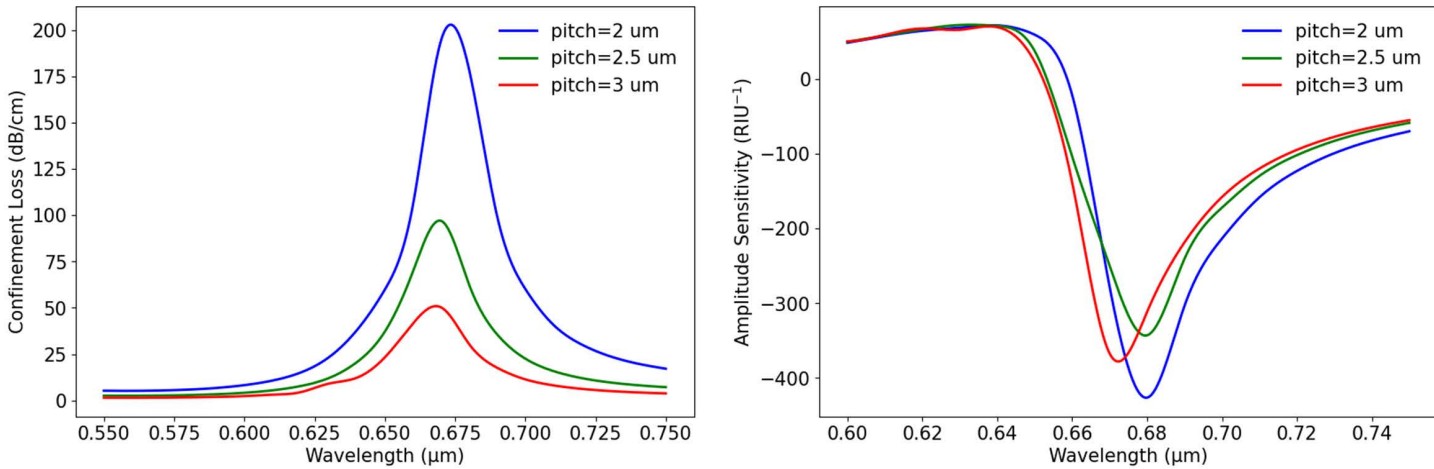

**Fig 9. (a) Wavelength vs. CL with variation of pitch (b) wavelength vs. $S_A$ based on pitch variation.**

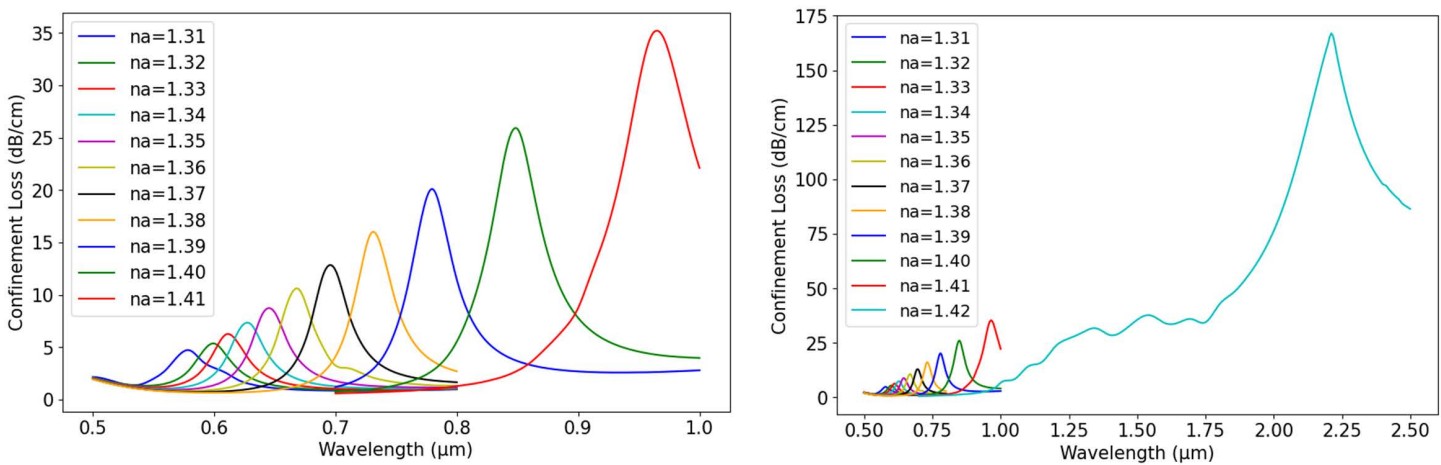

**Fig 10. CL vs. wavelength (a) $n_a$ = 1.31-1.41, (b) $n_a$ = 1.31-1.42.**

of CL on $n_a$, highlighting its importance in sensor performance. Fig 10(b) exhibits a similar pattern but extends the wavelength range up to 2.5 µm and includes an additional $n_a$ value of 1.42. As $n_a$ increases, the resonance peaks continue shifting toward longer wavelengths, with the highest CL observed for $n_a$ = 1.42. The peaks become more prominent, and loss values rise significantly, especially at higher $n_a$ values. These results emphasize the significant impact of $n_a$ on CL, showing that increasing $n_a$ shifts resonance to longer wavelengths while amplifying loss. This behaviour is critical for designing sensors with enhanced sensitivity and precision in detecting variations in $n_a$.

Again, Fig 11 displays how $S_A$ varies with wavelength across different analyte refractive indices, using fixed parameters: d1 = 0.6 µm, d2 = 1 µm, p = 3 µm, and a gold thickness of 60 nm. The variations in $S_A$ across different wl indicate the influence of $n_a$ on plasmonic interactions. As $n_a$ increases from 1.31 to 1.41, the sensitivity peaks shift towards longer wl. For $n_a$ = 1.31–1.32 (blue curve), the first significant drop occurs around 0.65 µm, whereas for $n_a$ = 1.40–1.41 (red curve), the minimum sensitivity shifts beyond 0.9 µm. The negative peaks indicate strong plasmonic coupling, highlighting the sensor's capability to detect small changes in $n_a$. These variations suggest that the optimal sensing wavelength shifts as

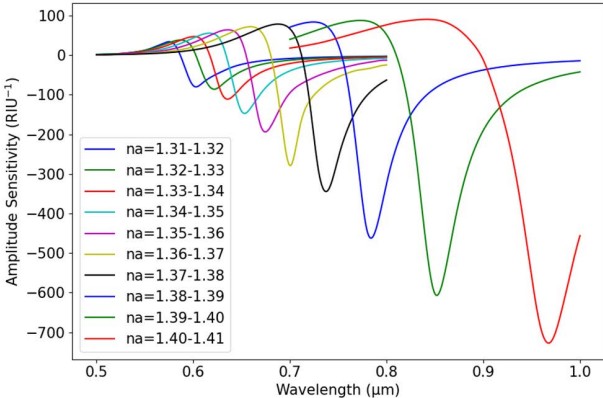

**Fig 11. $S_A$ vs. wavelength for $n_a = 1.31\text{-}1.41$.**

$n_a$ increases, making precise wl selection crucial for achieving high sensitivity. Overall, the results emphasize the impact of $n_a$ on $S_A$ and the importance of choosing the appropriate wl for effective sensing. The shift in resonance peaks with increasing $n_a$ highlights the need for wl tuning to maximize detection accuracy.

The Table 2 presents different characteristics of the SPR biosensor across various parameter values of $n_a$, with constant d1 = 0.6 μm, d2 = 1 μm, p = 3 μm, and $t_g$ = 60 nm. It presents CL, peak resonance wavelength ($\lambda_{peak}$), $S_A$, $S_\lambda$, resolution, and FOM corresponding to $n_a$ values. As $n_a$ increases, the resonance wavelength shifts to longer values. We observe that the peak loss found at 590 nm wavelength for $n_a = 1.31$, and it reaches 2210 nm for $n_a = 1.42$. $S_A$ remains low for smaller $n_a$ values but increases significantly beyond $n_a = 1.36$. At $n_a = 1.31$, $S_A$ is 10 (RIU$^{-1}$), but it jumps to 50 (RIU$^{-1}$) at $n_a = 1.38$ and reaches 1250 (RIU$^{-1}$) at $n_a = 1.41$, showing improved sensor response for higher na. $S_\lambda$ also increases as $n_a$ rises. It reaches a maximum of 125,000 nm/RIU at $n_a = 1.41$. The resolution improves with the increase of $n_a$, reaching $8.00 \times 10^{-7}$ RIU at $n_a = 1.41$, which employs the sensor's ability to detect small changes in $n_a$. Furthermore, FOM also rises steadily with the increase of $n_a$. It remains below 50 for lower $n_a$ values but increases beyond $n_a = 1.36$ and highest at 1402.28 at $n_a = 1.41$. These results show that the sensor performs improves with higher $n_a$ values, because CL, $S_A$, $S_\lambda$, and FOM reach their highest values.

**Table 2. Optical properties of the proposed SPR biosensor for different $n_a$.**

| $n_a$ | max_CL (dB/cm) | $\lambda_{peak}$ (nm) | $S_A$ (RIU$^{-1}$) | $S_\lambda$ (nm/RIU) | Resolution (RIU) | FOM (RIU$^{-1}$) |
|---|---|---|---|---|---|---|
| 1.31 | 4.62 | 590 | 3.23 | 1000 | $1.00 \times 10^{-5}$ | 20.99 |
| 1.32 | 5.36 | 600 | 14.98 | 1000 | $1.00 \times 10^{-5}$ | 23.28 |
| 1.33 | 6.23 | 610 | 35.69 | 2000 | $5.00 \times 10^{-5}$ | 48.61 |
| 1.34 | 7.23 | 630 | 29.14 | 2000 | $5.00 \times 10^{-5}$ | 49.27 |
| 1.35 | 8.32 | 650 | 35.85 | 2000 | $5.00 \times 10^{-5}$ | 48.68 |
| 1.36 | 10.52 | 670 | 60.03 | 3000 | $3.33 \times 10^{-5}$ | 76.82 |
| 1.37 | 12.36 | 700 | 66.80 | 3000 | $3.33 \times 10^{-5}$ | 73.07 |
| 1.38 | 15.96 | 730 | 82.40 | 5000 | $2.00 \times 10^{-5}$ | 122.66 |
| 1.39 | 20.10 | 780 | 86.27 | 7000 | $1.43 \times 10^{-5}$ | 157.23 |
| 1.4 | 25.84 | 850 | 89.32 | 11000 | $9.09 \times 10^{-6}$ | 253.56 |
| 1.41 | 34.64 | 960 | 90.92 | 125000 | $8.00 \times 10^{-7}$ | 1402.28 |
| 1.42 | 166.88 | 2210 | – | – | – | – |

Table 3 presents the performance evaluation of the PCF-SPR biosensor based on various design parameters and $n_a$. The design parameters are small and large air hole diameters (d1, d2), pitch (p), $t_g$, and the optical properties, such as $\lambda_{peak}$, CL, $S_\lambda$, resolution, FOM, and $S_A$. With the increase of $n_a$ from 1.31 to 1.41, the $\lambda_{peak}$ shifts to higher values that vary from 0.57 µm to 0.95 µm. Maximum CL varies significantly, reaching its highest value of 445.37 dB/cm at $n_a = 1.4$. Similarly, $S_\lambda$ increases and reaches a peak value of 105000 nm/RIU at $n_a = 1.41$. Resolution also improves as $n_a$ rises, reaching $9.52 \times 10^{-7}$ RIU at $n_a = 1.41$. Furthermore, the FOM shows a rapid increase and shows the highest value at 2112.15 RIU$^{-1}$ for $n_a = 1.41$, which confirms superior sensing capability. Again $S_A$ also rises with increasing $n_a$, where the highest value is 95.28 RIU$^{-1}$ at $n_a = 1.41$. These results suggest that the sensor performs with optimized parameters such as d1 = 0.6 µm, d2 = 1.0 µm, p = 2.5 µm, and $t_g = 40$ nm, contributing to maximum sensitivity and resolution. From this table, we get the results and insights with parameter variation on biosensor efficiency that help to select optimal design configurations for the best performing sensor.

Table 4 presents the maximum absolute $S_A$ values for different $n_a$ in a PCF-SPR biosensor. The listed parameters include the structural dimensions: d1, d2, p, $t_g$ along with the operating wavelength (wl). The $S_A$ (RIU$^{-1}$) values represent the sensor's ability to detect changes in the surrounding RI, making them crucial for evaluating the sensor's performance. As $n_a$ increases, $S_A$ values generally increase in magnitude, indicating enhanced sensitivity. The highest negative $S_A$ value of $-1422.34$ RIU$^{-1}$ is observed at $n_a = 1.39$, highlighting the sensor's peak detection capability at this $n_a$.

## 3.2 ML-based performance analysis

Using ML, we have used the simulation-based dataset to explore other design combinations and to explore better-performing designs. The dataset "Dataset 1" contains 5252 samples with eight features, where d1, d2, $t_g$, $n_a$, and wl serve as inputs, while $N_{eff}$, CL, and $S_A$ are designated as outputs. We have created two subsets, "Dataset 2" and "Dataset 3", from the initial primary "Dataset 1". "Dataset 2" contains the combined samples for $N_{eff}$ and CL, while "Dataset 3" includes samples for $S_A$. Using the train-test split method using the Scikit-learn library, we partitioned the dataset into training and testing sets with a 75% and 25% ratio. After preprocessing and splitting the data, each model was trained on the training set, and its performance was evaluated on the testing set. We implemented various ML algorithms employing ensemble techniques and subsequently applied XAI methods to these models to improve interpretability and transparency.

The polynomial equation $y = 40.15x^2 - 105.93x + 70$ shown in Fig 12 describes how the resonance wavelength shifts as $n_a$ changes. In this equation, x represents $n_a$, and y is the resonance wavelength in micrometers. The quadratic term ($40.15x^2$) means that the wavelength shift increases more rapidly as $n_a$ increases. The linear term ($-105.93x$) shows the direct impact of $n_a$ on resonance wavelength. The negative coefficient $-105.93$ means that for smaller values of $n_a$ the

**Table 3. Optical properties for different $n_a$ with best performing design parameters.**

| d1 (µm) | d2 (µm) | p (µm) | $t_g$ (nm) | $n_a$ | $\lambda_{peak}$ (µm) | max_CL (dB/cm) | $S_\lambda$ (nm/RIU) | Resolution (RIU) | FOM (RIU$^{-1}$) | $S_A$ (RIU$^{-1}$) |
|---|---|---|---|---|---|---|---|---|---|---|
| 0.6 | 1.0 | 2.0 | 50 | 1.31 | 0.58 | 28.81 | 1000 | $1.00 \times 10^{-4}$ | 28.85 | 22.00 |
| 0.6 | 1.0 | 3.0 | 50 | 1.32 | 0.58 | 8.11 | 2000 | $5.00 \times 10^{-5}$ | 48.88 | 27.85 |
| 1.2 | 0.6 | 2.0 | 40 | 1.33 | 0.59 | 98.99 | 2000 | $5.00 \times 10^{-5}$ | 53.54 | 35.59 |
| 0.6 | 1.0 | 3.0 | 40 | 1.34 | 0.6 | 24.04 | 2000 | $5.00 \times 10^{-5}$ | 56.57 | 71.91 |
| 1.2 | 0.6 | 2.0 | 40 | 1.35 | 0.62 | 152.20 | 3000 | $3.33 \times 10^{-5}$ | 90.59 | 64.16 |
| 1.6 | 0.8 | 2.0 | 40 | 1.36 | 0.65 | 230.99 | 3000 | $3.33 \times 10^{-5}$ | 101.45 | 68.54 |
| 1.6 | 0.8 | 2.0 | 40 | 1.37 | 0.68 | 302.80 | 4000 | $2.50 \times 10^{-5}$ | 141.23 | 78.54 |
| 1.6 | 0.6 | 2.0 | 40 | 1.38 | 0.71 | 437.15 | 6000 | $1.67 \times 10^{-5}$ | 250.99 | 89.28 |
| 0.6 | 1.0 | 2.0 | 40 | 1.39 | 0.77 | 372.63 | 10000 | $1.00 \times 10^{-5}$ | 338.63 | 71.04 |
| 0.6 | 1.0 | 2.0 | 40 | 1.4 | 0.86 | 445.37 | 87000 | $1.15 \times 10^{-6}$ | 1718.34 | 88.37 |
| 0.6 | 1.0 | 2.5 | 40 | 1.41 | 0.95 | 333.97 | 105000 | $9.52 \times 10^{-7}$ | 2112.15 | 95.28 |

Table 4.  Maximum absolute $S_A$ values for different $n_a$.

| d1(µm) | d2(µm) | p(µm) | $t_g$(nm) | $n_a$ | wl(µm) | $S_A$(RIU$^{-1}$) |
|---|---|---|---|---|---|---|
| 1.6 | 0.8 | 2.0 | 40 | 1.31 | 0.59 | −82.11 |
| 1.6 | 0.8 | 2.0 | 40 | 1.32 | 0.60 | −107.04 |
| 1.6 | 0.8 | 2.0 | 40 | 1.33 | 0.62 | −140.07 |
| 1.6 | 0.8 | 2.0 | 40 | 1.34 | 0.63 | −205.60 |
| 1.6 | 0.8 | 2.0 | 40 | 1.35 | 0.65 | −300.03 |
| 1.6 | 0.8 | 2.0 | 40 | 1.36 | 0.68 | −481.88 |
| 1.6 | 0.8 | 2.0 | 40 | 1.37 | 0.72 | −721.13 |
| 1.2 | 0.8 | 2.0 | 40 | 1.38 | 0.77 | −1209.86 |
| 0.6 | 1.0 | 3.0 | 30 | 1.39 | 0.74 | −1422.34 |
| 0.6 | 1.0 | 2.0 | 30 | 1.40 | 1.00 | −505.40 |
| 0.6 | 1.0 | 2.5 | 40 | 1.41 | 0.95 | 95.28 |

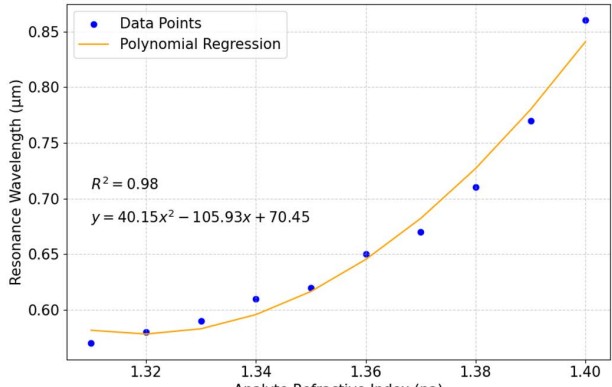

**Fig 12.  Polynomial regression analysis of resonance wavelength with the change of $n_a$.**

resonance wavelength (µm) initially decreases. However, as $n_a$ increases, the positive quadratic term $40.15x^2$ starts to dominate linear term ($−105.93x$), causing the resonance wavelength to rise sharply. The constant term (70.45) is the estimated resonance wavelength when $n_a$ is zero, though this value is theoretical and does not hold physical significance. The equation follows a curved trend rather than a straight-line change, meaning that at higher $n_a$ values, small shifts cause more noticeable wavelength changes. This pattern is useful in biosensor applications, where detecting small $n_a$ variations is important with high sensitivity and accuracy.

Fig 13 compares the actual and predicted $N_{eff}$ values using the RFR model. The actual values are marked in purple, while the predicted values are in orange. The graph shows that the RFR model predicts $N_{eff}$ accurately, with minor deviations in some areas. Additionally, Fig 14 presents the validation of different ML models for predicting $N_{eff}$. We have plotted together the actual data alongside the predicted data using the DTR, RFR, GBR, XGBR, and BR models. The close match between actual and predicted values implies that all models perform well, showing minimal variations. The validation confirms that these models generalize effectively, maintaining consistency across different wavelength values.

Table 5 provides performance metrics such as R², MAE, and MSE for both training and test datasets for different ML models. Here, DTR achieves the highest train R² (0.999969) but has slightly lower test accuracy. RFR, BR, and XGBR show strong generalization, with high R² values and low errors. KNN records the lowest test R² (0.992828) and the

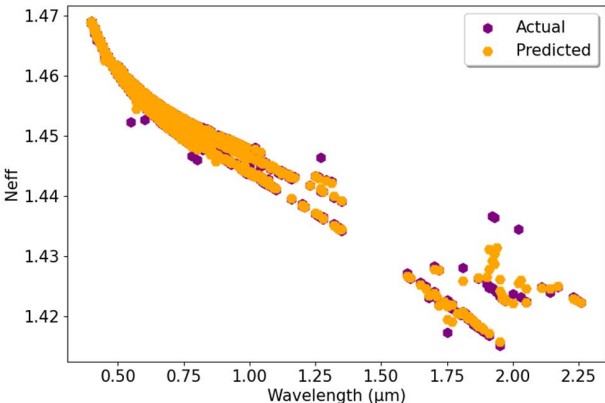

**Fig 13. Actual vs. predicted N$_{eff}$ using RFR model.**

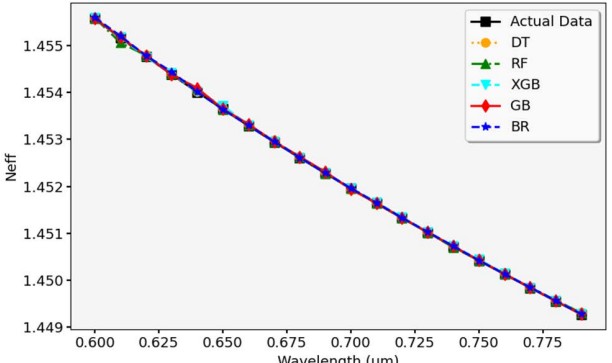

**Fig 14. Validation of N$_{eff}$ predictions across different ML models.**

highest test MAE. (0.007912), indicating lower accuracy. GBR performs well, achieving a high test R² (0.997951) and the lowest test MSE (0.000053). The validation results suggest that ensemble models such as RFR, GBR, and XGBR provide the most reliable predictions for N$_{eff}$ with minimal error. In terms of execution time, the DTR is the fastest, with the lowest training time (0.020 sec) and test time (0.000099 sec). However, it is less accurate compared to ensemble models. While XGBR offers better efficiency, with a lower training time (0.238 sec) compared to RFR (0.618 sec), RFR remains a strong performer due to its consistent performance. Although GBR is slightly more accurate, it requires more computational resources. Overall, XGBR and RFR emerge as the best performers, offering strong predictive accuracy with a balance of execution efficiency.

Furthermore, we have calculated the SHAP values using the RFR model, to assess feature importance and interpretability in the analysis. SHAP values quantify how much each feature of the input data influences the model's prediction, relative to the average prediction. Positive SHAP values indicate an increase in the target value N$_{eff}$, while negative values mean a decrease. It also helps to understand how much each input attribute contributes to N$_{eff}$'s final prediction. Fig 15 illustrates this by combining feature importance with feature effects. Each point on the summary plot represents a Shapley value for a feature and an instance, with the y-axis indicating the feature and the x-axis the Shapley value. The color gradient from blue to red represents the feature's value from low to high. Notably, wl has a significant impact on N$_{eff}$, with higher values (in red) decreasing and lower values (in blue) increasing. This also shows that wl has a significant negative

**Table 5. Comparative performance metrics for $N_{eff}$ across different ML models.**

| Models | Train $R^2$ | Test $R^2$ | Train MAE | Test MAE | Train MSE | Test MSE | Execution Time (sec) | |
|---|---|---|---|---|---|---|---|---|
| | | | | | | | Train time | Test time |
| DTR | 0.999969 | 0.996479 | $4.50 \times 10^{-5}$ | $2.34 \times 10^{-3}$ | $8.10 \times 10^{-7}$ | $8.90 \times 10^{-5}$ | 0.020045 | 0.000099 |
| RFR | 0.999592 | 0.997477 | $8.44 \times 10^{-4}$ | $2.22 \times 10^{-3}$ | $1.07 \times 10^{-5}$ | $6.50 \times 10^{-5}$ | 0.617616 | 0.007231 |
| KNN | 0.996601 | 0.992828 | $4.35 \times 10^{-3}$ | $7.91 \times 10^{-3}$ | $8.95 \times 10^{-5}$ | $1.86 \times 10^{-4}$ | 0.048177 | 0.008654 |
| GBR | 0.999261 | 0.997951 | $2.16 \times 10^{-3}$ | $2.71 \times 10^{-3}$ | $1.95 \times 10^{-5}$ | $5.30 \times 10^{-5}$ | 0.387928 | 0.008544 |
| XGBR | 0.999401 | 0.997684 | $1.52 \times 10^{-3}$ | $2.22 \times 10^{-3}$ | $1.58 \times 10^{-5}$ | $5.90 \times 10^{-5}$ | 0.237774 | 0.007392 |
| BR | 0.999593 | 0.997447 | $8.43 \times 10^{-4}$ | $2.22 \times 10^{-3}$ | $1.07 \times 10^{-5}$ | $6.60 \times 10^{-5}$ | 0.614761 | 0.006754 |

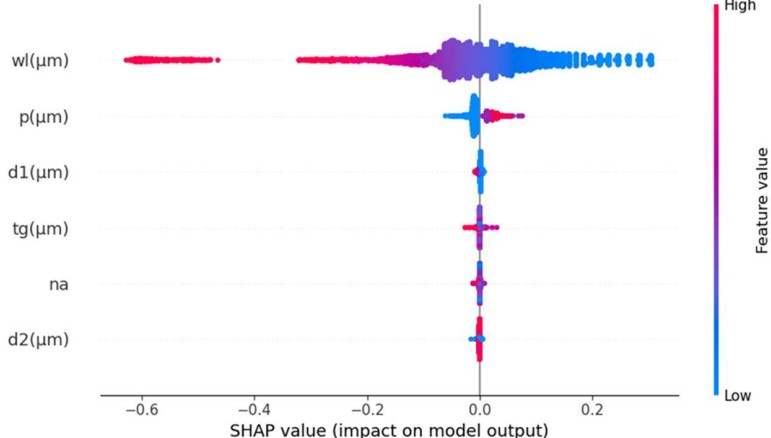

**Fig 15. SHAP summary plot for $N_{eff}$ prediction.**

impact on $N_{eff}$, underscoring its crucial role in determining $N_{eff}$, while other variables have a less significant influence on $N_{eff}$ value prediction.

Again, the SHAP waterfall plot in Fig 16 shows how different input features contribute to predicting $N_{eff}$, starting from a base value E[f(x)] = 1.453 and going to a final predicted value f(x) = 1.454. The plot depicts that each input feature either increases or decreases $N_{eff}$, impacting the final predicted value. The blue bars represent negative contributions, while the red bars indicate positive contributions. It reveals that wavelength (wl = 0.63 μm) has the most significant impact, which increases $N_{eff}$ by 0.00187. Pitch (p = 2 μm) decreases $N_{eff}$ by 0.00064, while d1 (1.2 μm) slightly decreases it by 0.00012. The influence of d2 (1 μm) is a small negative contribution of 0.00008. The $n_a$ = 1.35 has a minor positive effect of 0.00003, whereas the $t_g$ = 0.04 μm has no noticeable impact. This SHAP plot provides insight into how input parameters influence the predicted $N_{eff}$ value. We observe that wl is the most dominant factor, followed by p, while other features have a much smaller effect on the final prediction.

**3.2.1 Confinement loss (CL).** The graph in Fig 17 shows how the predicted CL values compare to the actual ones using the RFR model. The actual data points, represented in purple, are plotted alongside the predicted values in orange. The pattern suggests that the model accurately follows the trend of CL, although slight differences are noticeable in higher ranges. Overall, the predictions align well with actual values across different wavelengths, demonstrating the model's effectiveness. A further evaluation of ML models in predicting CL is displayed in Fig 18. The actual data is plotted alongside predicted values using DTR, RFR, GBR, XGBR, and BR models. The results indicate that most of the models match the actual data closely, showing minimal variation, which confirms their reliability for generalization.

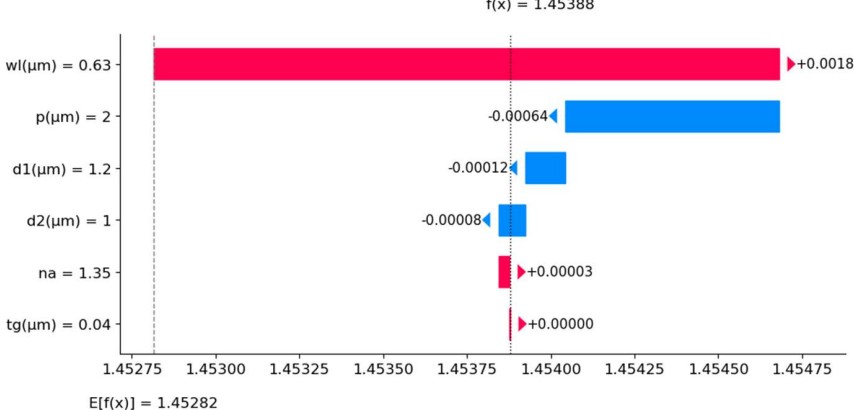

**Fig 16. SHAP waterfall plot for $N_{eff}$ prediction.**

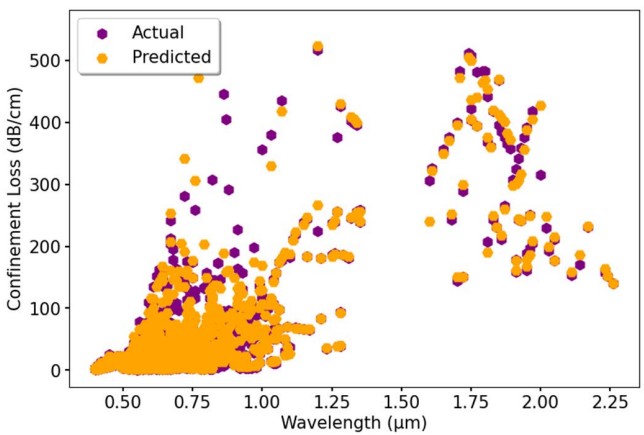

**Fig 17. Actual vs. predicted CL using RFR model.**

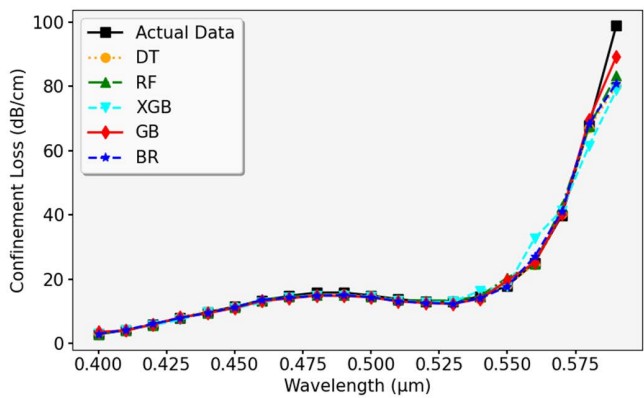

**Fig 18. Validation of different ML models for CL prediction against actual data.**

Additionally, to analyze performance, Table 6 provides a comparison of performance metrics such as R², MAE, and MSE for both training and test datasets. Where DTR has a high train R² of 0.999444, its lower test R² of 0.927159 suggests reduced accuracy when handling new data. RFR and BR exhibit strong predictive capabilities, with test R² values of 0.96148 and 0.961493, respectively, along with lower error rates. XGBR also performs well, achieving a test R² of 0.941124 and a test MSE of 0.000935. KNN and GBR show slightly lower test R² values (0.909516 and 0.912935) and higher MAE, indicating relatively lower accuracy compared to ensemble models. For CL prediction, DTR is the fastest, with a training time of 0.021 seconds and a test time of 0.0003 seconds, but it has lower test accuracy. RFR and BR achieve the highest test R² but require longer training times (0.627 and 0.655 seconds). XGBR provides a strong balance, with a high test R², a low test MSE, and moderate training time (0.261 seconds). KNN and GBR have the lowest test R² and higher errors, making them less suitable. Among all, RFR and BR demonstrate the most reliable CL predictions with minimal error.

Like $N_{eff}$ analysis, we also used SHAP to explore the impact of different input parameters on CL. SHAP values were computed using the RFR model that is one of the most effective models in this study. The summary plot in Fig 19 presents the distribution of SHAP values for various features for CL prediction. Each point represents an individual prediction instance, with the y-axis displaying the features and the x-axis showing their corresponding SHAP values. The color gradient from blue to red reflects the feature values, where blue denotes lower values and red represents higher ones. Among all parameters, wl emerges as the most influential, showing both high and low values leading to significant variations in

**Table 6. Performance metrics of ML models for CL prediction.**

| Models | Train R² | Test R² | Train MAE | Test MAE | Train MSE | Test MSE | Execution Time (sec) | |
|---|---|---|---|---|---|---|---|---|
| | | | | | | | Train time | Test time |
| DTR | 0.999444 | 0.927159 | $1.64 \times 10^{-4}$ | $1.01 \times 10^{-2}$ | $9.00 \times 10^{-6}$ | $1.16 \times 10^{-3}$ | 0.021194 | 0.000303 |
| RFR | 0.993567 | 0.96148 | $3.37 \times 10^{-3}$ | $8.58 \times 10^{-3}$ | $1.03 \times 10^{-4}$ | $6.22 \times 10^{-4}$ | 0.62719 | 0.010593 |
| KNN | 0.953331 | 0.909516 | $9.81 \times 10^{-3}$ | $1.45 \times 10^{-2}$ | $7.49 \times 10^{-4}$ | $1.43 \times 10^{-3}$ | 0.05 | 0.005948 |
| GBR | 0.946968 | 0.912935 | $1.29 \times 10^{-2}$ | $1.62 \times 10^{-2}$ | $8.50 \times 10^{-4}$ | $1.40 \times 10^{-3}$ | 0.423946 | 0.005308 |
| XGBR | 0.976157 | 0.941124 | $8.68 \times 10^{-3}$ | $1.29 \times 10^{-2}$ | $3.83 \times 10^{-4}$ | $9.35 \times 10^{-4}$ | 0.261034 | 0.004159 |
| BR | 0.993551 | 0.961493 | $3.37 \times 10^{-3}$ | $8.57 \times 10^{-3}$ | $1.04 \times 10^{-4}$ | $6.22 \times 10^{-4}$ | 0.654856 | 0.011571 |

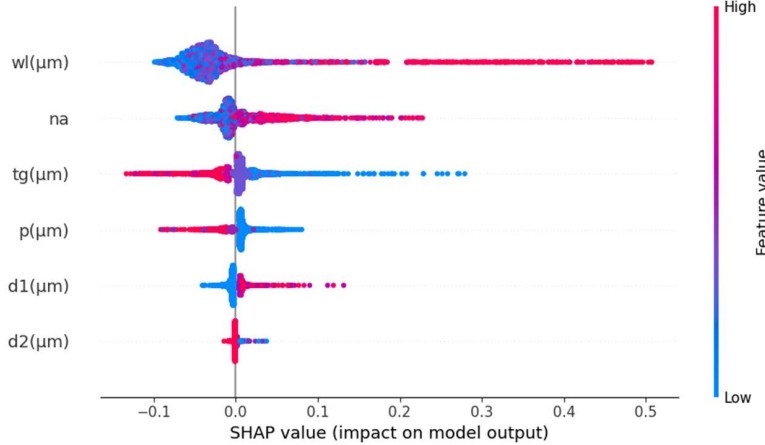

**Fig 19. SHAP summary plot for CL prediction.**

the model's output. The $n_a$ and $t_g$ also play a crucial role, generally increasing CL with higher values. Other parameters, such as p, d1, and the large air hole d2, have a smaller but still noticeable effect. The variations in SHAP values suggest complex feature interactions, which are essential for refining the model's interpretability. A SHAP waterfall plot in Fig 20 shows the change from the baseline E[f(X)] =47.157 to the final prediction f(x)= 77.28. This shows how these features have an effect. The $n_a$ parameter makes the most substantial contribution, increasing CL by 45.61283 units. In contrast, wl reduces the prediction by −35.02253 units, while $t_g$ contributes an additional 21.55977 units. Similarly, d2 leads to a decrease of 2.51656 units, whereas p has a minimal positive and d2 negative impact. These changes demonstrate how different input variables influence CL, with wavelength standing out as the dominant factor.

**3.2.2 Amplitude sensitivity ($S_A$).** In Fig 21, actual and predicted $S_A$ values are compared, where actual values are shown in purple and predicted values are marked in orange. The results indicate that the model has excellent predictions, but some deviations appear, particularly at lower sensitivity values. Fig 22 presents a detailed evaluation to predict $S_A$ using ML models. The black line represents the actual data, while various colors display the predictions from DTR, RFR, XGBR, GBR, and BR. Most models follow the expected trend, but RFR and BR exhibit the closest match to actual values, indicating their strong predictive accuracy.

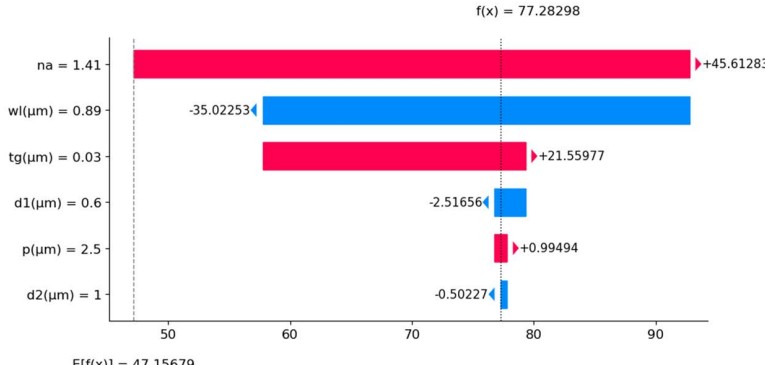

**Fig 20. SHAP waterfall plot for CL prediction.**

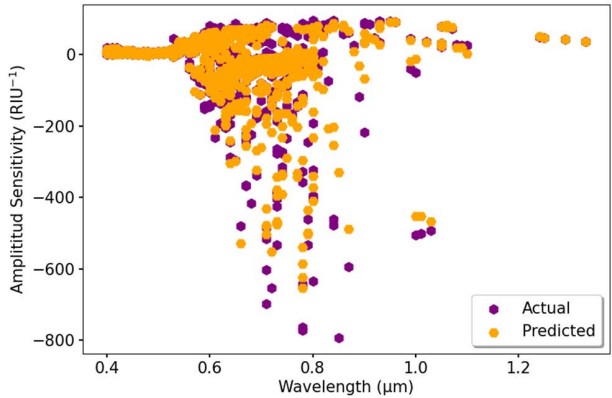

**Fig 21. Actual vs. predicted result of $S_A$ using RFR model.**

 

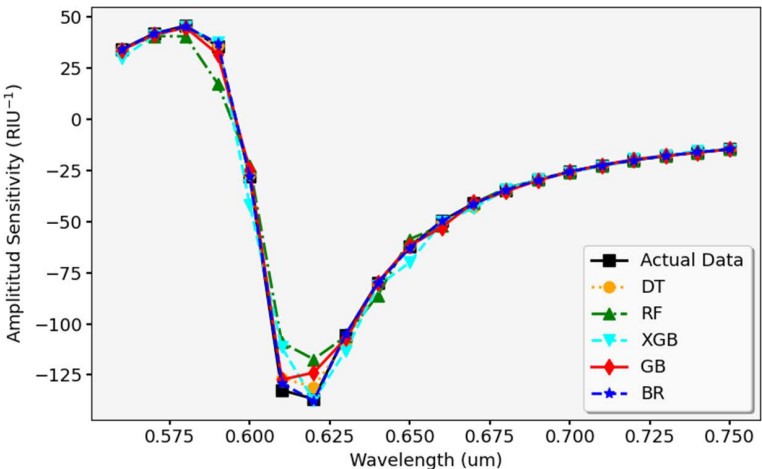

**Fig 22. Validation of S$_A$ predictions from various ML models against experimental data.**

Again, Table 7 presents key performance metrics such as R², MAE, and MSE for both training and test datasets for predicting S$_A$. Where, DTR shows an R² of 1.0 in training but drops to 0.788216 in testing, indicating overfitting. Its test MAE (0.010117) and MSE (0.001413) are relatively high, suggesting weaker generalization. DTR is the fastest, with a training time of 0.018 seconds and a test time of 0.0005 seconds, but has a lower test R² (0.788216). RFR and BR achieve the highest test R² (0.868116 and 0.867602) with a test MAE around 0.00924 but require longer training times (0.474 and 0.470 seconds). XGBR provides a strong balance, with a test R² of 0.813183, low test MSE (0.001356), and moderate training time (0.218 seconds). KNN and GBR perform weaker, with KNN showing the lowest test R² (0.50024) and highest test MAE (0.023273). Overall, RFR and BR offer the best accuracy, their balance of high accuracy and low error makes them well-suited for S$_A$ prediction.

The SHAP summary plot in Fig 23 makes it very clear that the feature wl has the biggest effect on S$_A$. This highlights how important it is for optimal sensor model performance. Additionally, the feature also demonstrates a substantial impact, underscoring its importance in predicting S$_A$. The SHAP summary plot also depicts that t$_g$ has a moderate effect, indicating its notable impact on predictions. In contrast, d1 and d2 show minimal impacts, suggesting that these parameters have less influence on S$_A$ compared to wl, n$_a$, and t$_g$. Again, Fig 24 illustrates a SHAP waterfall model with a specific s$_a$mple. It shows how various features contribute to the model's final prediction, starting with a baseline value of E [f(X)] = −32.067 and reaching an output of f(x) = −36.866. Here, the feature wl has the most significant negative impact, decreases the prediction by −9.11925 units. The n$_a$ feature also adds a substantial 2.01146 units to the prediction. In contrast, the feature

**Table 7. S$_A$ Prediction performance metrics for different ML models.**

| Models | Train R² | Test R² | Train MAE | Test MAE | Train MSE | Test MSE | Execution Time (sec) | |
|---|---|---|---|---|---|---|---|---|
| | | | | | | | Train time | Test time |
| DTR | 1 | 0.788216 | $9.24 \times 10^{-7}$ | 0.010117 | $1.54 \times 10^{-9}$ | 0.001413 | 0.017517 | 0.000493 |
| RFR | 0.977498 | 0.868116 | $3.53 \times 10^{-3}$ | 0.00923 | $1.57 \times 10^{-4}$ | 0.000985 | 0.474224 | 0.004532 |
| KNN | 0.747129 | 0.50024 | $1.54 \times 10^{-2}$ | 0.023273 | $1.76 \times 10^{-3}$ | 0.003449 | 0.036002 | 0.004144 |
| GBR | 0.917111 | 0.789826 | $9.54 \times 10^{-3}$ | 0.014027 | $5.74 \times 10^{-4}$ | 0.001557 | 0.311306 | 0.003934 |
| XGBR | 0.949936 | 0.813183 | $8.25 \times 10^{-3}$ | 0.013992 | $3.48 \times 10^{-4}$ | 0.001356 | 0.218452 | 0.003612 |
| BR | 0.977638 | 0.867602 | $3.52 \times 10^{-3}$ | 0.009254 | $1.56 \times 10^{-4}$ | 0.000988 | 0.470452 | 0.008843 |

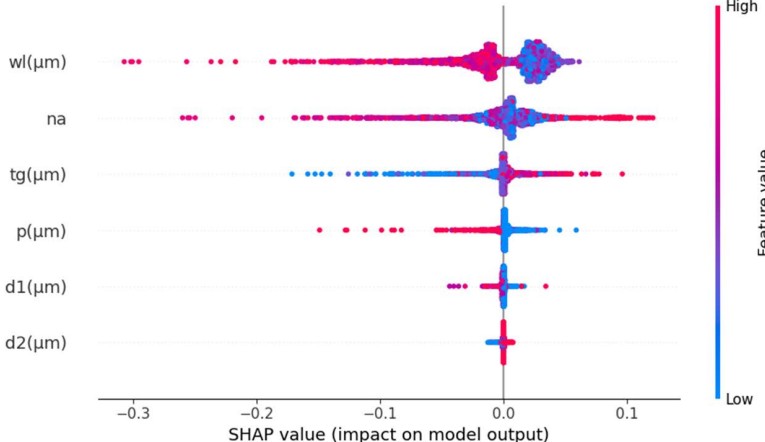

**Fig 23. SHAP summary plot of S$_A$ prediction.**

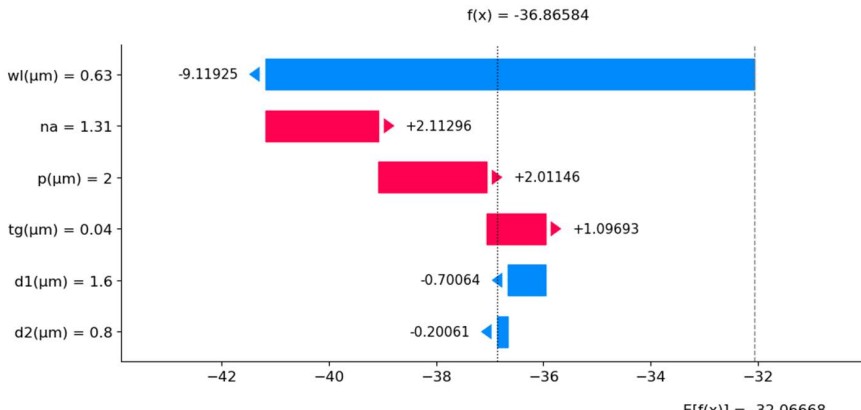

**Fig 24. SHAP waterfall plot of S$_A$ prediction.**

$t_g$ increases the prediction by 1.09693 units. The d1 and d2 features contribute small negative adjustments. This demonstrates the sensitivity of the model's output to changes in each feature, with wl and $n_a$ being the most influential in driving the prediction upward.

### 3.3 Comparison with previous works

Table 8 compares the performance of our proposed PCF-SPR biosensor with recent works, highlighting significant improvements in optical properties. The proposed design outperforms previous models in terms of CL, S$_A$, S$_\lambda$, resolution, and FOM, demonstrating superior sensing capabilities. In terms of S$_\lambda$, the proposed biosensor achieves 125,000nm/RIU, which is higher than all the works. The S$_A$ of −1422.34 RIU$^{-1}$ remains competitive, close to 7220 RIU$^{-1}$ [46] and −1971.30 RIU$^{-1}$ [47], while excelling in other performance areas. Additionally, the proposed biosensor achieves an exceptionally low resolution of 8×10$^{-7}$ RIU, which is superior to previous models [20,21,47–52]. Among all the properties, the FOM of 2112.15 RIU$^{-1}$ is significantly higher than those reported in other studies [17,22,49,50,52] indicating a better balance between sensitivity and resolution. In our study, we employed six ML models (DTR, RFR, KNN, GBR, XGBR, BR), along

**Table 8. Comparison with previous work.**

| References | $n_a$ Range | Wavelength Range (nm) | Maximum $S_\lambda$ (nm/RIU) | Maximum $S_A$ ($RIU^{-1}$) | R (RIU) | FOM ($RIU^{-1}$) | ML Application | |
|---|---|---|---|---|---|---|---|---|
| | | | | | | | Model | Best Performance |
| [17](2024) | 1.33 - 1.43 | 550-3500 | 123,000 | – | $8.13 \times 10^{-7}$ | 683 | ANN | MSE = 0.0097 $R^2$ = 0.9987 |
| [21](2024) | 1.31 to 1.40 | 720-1280 | 18000 | 889.89 | $5.56 \times 10^{-6}$ | – | ANN | MSE = 0.002 |
| [22](2024) | 1.380−.401 | 960-1060 | 13,257.20 | – | – | 36.52 | – | – |
| [48](2024) | 1.36-1.38 | 600-1000 | 7143 | −270 | $2.9 \times 10^{-5}$ | – | – | – |
| [46](2023) | 1.26-1.36 | 1500–2000 | 48,000 | 7220 | – | – | DNN, GB | $R^2$ = 0.98, MAE = 0.007 |
| [47](2024) | 1.362-1.401 | 600-1600 | 35,714.28 | −1971.30 | $2.80 \times 10^{-6}$ | – | – | – |
| [20](2024) | 1.37 - 1.41 | 600-900 | 5500 | – | $2.05 \times 10^{-5}$ | – | SVM | $R^2$ = 0.96 F1-Score = 0.958 MCC = 0.923 |
| [49](2024) | 1.27 - 1.41 | 539-900 | 12,300 | 1623.6 | $8.13 \times 10^{-6}$ | 560 | – | – |
| [50](2024) | 1.25–1.36 | 1120–2050 | 33,000 | – | $3.03 \times 10^{-6}$ | 268.29 | – | – |
| [51](2024) | 1.28 - 1.44 | 740-1000 | 1000 | 98.422 | 0.001 | – | MLR, ANN | $R^2$ = 0.98 RMSE = 0.1647 $R^2$ = 0.99 RMSE = 0.1585 |
| [52](2025) | 1.26-1.38 | 692-1593 | 5400 | – | $3.2 \times 10^{-5}$ | 120.43 | – | – |
| proposed | 1.31-1.42 | 400-2300 | 125,000 | −1422.34 | $8 \times 10^{-7}$ | 2112.15 | DTR, RFR, KNN, GB, XGBR, BR, XAI | $R^2$ = 0.9975 MAE = 0.0022 MSE = 0.00059 |

with XAI techniques, specifically SHAP, to gain insights into the model's decision-making process. The XGBR model demonstrated excellent performance with an R² value of 0.9976 for the prediction of $N_{eff}$, outperforming other models in the literature. The use of XAI methods like SHAP made the results more interpretable, providing valuable information about how different design parameters impact the sensor's performance. This transparency not only enhanced our understanding of the system but also allowed for more informed decisions during the optimization process. The ML models showed lower error rates and higher accuracy compared to previous works, achieving an MAE of 0.00221 and an MSE of 0.00059, which are significantly lower than most of the other studies. This demonstrates the robustness of the ML models in predicting sensor performance. Additionally, integrating XAI with ML optimization significantly reduced computational time and cost, making it more efficient than conventional methods for exploring optimal design combinations. Overall, the proposed PCF-SPR biosensor, with its combination of ML and XAI techniques, shows substantial improvements in $S_A$, $S_\lambda$, resolution, and FOM compared to previous designs. This makes it a promising candidate for high-precision applications in various sensing domains, while the inclusion of XAI offers valuable insights into the underlying sensor behavior.

Furthermore, compared to traditional PCF-SPR sensor optimization approaches that rely solely on parametric sweeps using FEM, our hybrid methodology incorporating ML (e.g., RFR, XGB) and SHAP-based XAI significantly enhances computational efficiency and interpretability. FEM alone often demands substantial time and computing power for iterative tuning of design parameters. In contrast, our work reduces the time for predicting key optical parameters such as $N_{eff}$, CL, and $S_A$ from minutes to milliseconds, enabling rapid exploration of a broader design space. Recent sensor architectures have demonstrated high performance using complex geometries. Such as IMD-EMD merged PCF [53] and quasi D-shaped waveguides with external metal deposition [54], offering high sensitivities for detecting heavy metals, oils [55], and cancer biomarkers [56]. However, these structures typically require higher fabrication and simulation costs. Our proposed sensor, in comparison, achieves high performance metrics ($S_\lambda$ = 125,000 nm/RIU and FOM = 2112.15) using a simpler geometry with the help of an intelligent data-driven approach. By combining ML and XAI with efficient sensor design, this framework supports rapid development of adaptive, intelligent biosensing platforms for diverse real-world applications.

## 4. Discussion

Moreover, from our results and analysis, it confirms that ML models can accurately predict various optical properties. By providing rapid and precise predictions, ML lowers computational costs and time and accelerates the optimization process for high-performance PCF-SPR biosensor design. This approach enables efficient exploration of the best-performing configurations by evaluating numerous design variations without additional simulations. Furthermore, the comparison between COMSOL simulations and ML models highlights the significant advantage of ML in terms of computational efficiency. While COMSOL requires approximately 2 minutes for each single-mode calculation, ML models achieve similar predictions in only milliseconds. Among the tested RFR and XGBoost, RFR provides the best balance between accuracy and execution time. The drastic reduction in computation time makes ML-based approaches highly efficient for rapid mode prediction, offering a promising alternative for scenarios requiring multiple evaluations with minimal processing overhead.

Additionally, SHAP analysis provides interpretability and a clear understanding of each input feature's influence on the sensor's optical properties. This work effectively enhanced the sensor design by using COMSOL simulations to create data, ML for rapid optimization, and SHAP for better understanding. The SHAP analysis reveals that Au thickness ($t_g$) and pitch (p) play notable roles in sensor performance. For $N_{eff}$, p has a moderate negative effect, indicating that increasing pitch slightly lowers the effective index due to reduced core confinement. In contrast, $t_g$ shows minimal influence on $N_{eff}$. For CL, $t_g$ significantly increases loss, aligning with its known effect of enhancing plasmonic interaction but introducing absorption. Pitch has a minor positive effect on CL, suggesting a marginal loss increase with wider spacing. In $S_A$ prediction, $t_g$ moderately boosts sensitivity, confirming its importance in optimizing SPR strength. However, p has minimal effect on $S_A$. Overall, $t_g$ is more critical for optimizing $S_A$ and CL, while pitch subtly influences mode confinement.

The performance of the proposed PCF-SPR biosensor is evaluated using two critical metrics, $S_\lambda$ and FOM, which together determine the sensor's effectiveness in high-precision detection tasks. The sensor achieves exceptional values, with a maximum $S_\lambda$ of 125,000 nm/RIU and FOM of 2112.15, indicating its strong capability to detect change of $n_a$. High $S_\lambda$ ensures that even minimal variations in the $n_a$ cause significant shifts in the resonance wavelength, making the sensor highly effective for detecting biochemical interactions. The FOM, which combines sensitivity with spectral resolution, reflects the sharpness of the resonance peak, an essential factor for achieving accurate and low-error detection results.

Such sensor performance is particularly advantageous in medical diagnostics, especially for detecting cancer biomarkers [33,56], where early-stage detection requires identifying minute biomolecular changes in samples like blood or serum. The integration of ML and XAI into the design process enables accurate prediction of optical responses and efficient parameter optimization. This not only reduces reliance on computationally intensive simulations but also enhances the sensor's adaptability in real-time, making it well-suited for dynamic and complex biological systems. Our proposed design, covering a wide $n_a$ detection range of 1.31–1.42, allows for effective operation across diverse diagnostic media. Beyond medical diagnostics, the biosensor's high sensitivity, low CL, and wide operational range make it equally applicable to chemical and environmental sensing tasks, such as the detection of organic solvents, contaminants, and pharmaceutical compounds.

While ML models are highly effective at predicting key optical properties such as $N_{eff}$, CL, and sensitivity, certain non-quantifiable or dynamic sensing parameters remain challenging to predict. For instance, parameters influenced by fabrication tolerances, material degradation, surface roughness, and temperature-dependent shifts are often difficult to capture accurately using ML unless real experimental data are integrated into the training process. Similarly, nonlinear environmental effects, such as biofouling or long-term drift, typically require real-time adaptive models or physical testing for accurate assessment. Despite these limitations, the ML-based predictions offer significant benefits for the overall sensor design process. They drastically reduce computational load, enable rapid evaluation of large design spaces, and help in identifying optimal configurations without exhaustive simulation runs. More importantly, explainable ML techniques such as SHAP add interpretability, allowing designers to understand which parameters most strongly influence sensor behavior.

## 5. Conclusion

This study demonstrates the significant enhancement of optical properties in PCF-SPR biosensors by integrating ML with simulation-based design. The proposed biosensor features a simple design with low CL, high $S_A$, $S_\lambda$, and a high FOM. Operating over $n_a$ range from 1.31 to 1.42, with gold employed as the plasmonic material. The biosensor achieves exceptional performance metrics, including a maximum $S_\lambda$ of 125,000 nm/RIU, an $S_A$ of −1422.34 RIU$^{-1}$, a resolution of $8 \times 10^{-7}$ RIU, and a FOM of 2112.15. The integration of various ML models, including RFR, DTR, GBR, XGBR, KNN, and BR, accelerated the optimization process, significantly reducing computational costs and time compared to conventional methods. The XGBR model demonstrated excellent predictive accuracy ($R^2 = 0.9976$, MAE = 0.00221, MSE = 0.00059) for $N_{eff}$ predictions. Specially, the use of XAI techniques, particularly SHAP, provided valuable insights into how design parameters influence sensor performance, enabling efficient exploration of optimal parameter combinations. Due to its high sensitivity and low CL, the proposed biosensor is well-suited for detecting small $n_a$ variations, making it ideal for medical diagnostics, disease detection, environmental monitoring, biochemical sensing, and industrial applications. Future work can focus on exploring alternative plasmonic materials, enhancing the generalization of ML models, and conducting experimental validations to further improve the sensor's reliability and real-world applicability.

## Author contributions

**Conceptualization:** Mst. Rokeya Khatun, Md. Saiful Islam.

**Data curation:** Mst. Rokeya Khatun.

**Formal analysis:** Mst. Rokeya Khatun, Md. Saiful Islam.

**Investigation:** Mst. Rokeya Khatun.

**Methodology:** Mst. Rokeya Khatun.

**Software:** Mst. Rokeya Khatun.

**Supervision:** Md. Saiful Islam.

**Validation:** Mst. Rokeya Khatun.

**Visualization:** Mst. Rokeya Khatun.

**Writing – original draft:** Mst. Rokeya Khatun.

**Writing – review & editing:** Md. Saiful Islam.

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
