## [Decision Letter · Decision Letter 0]

18 Jun 2025

Dear Dr. Khatun,

Thank you for submitting your manuscript to PLOS ONE. After careful consideration, we feel that it has merit but does not fully meet PLOS ONE’s publication criteria as it currently stands. Therefore, we invite you to submit a revised version of the manuscript that addresses the points raised during the review process.

We look forward to receiving your revised manuscript.

Kind regards,

Yuan-Fong Chou Chau

Academic Editor

PLOS ONE

Reviewers' comments:

Reviewer's Responses to Questions

**Comments to the Author**

1. Is the manuscript technically sound, and do the data support the conclusions?

Reviewer #1: Yes

Reviewer #2: Yes

2. Has the statistical analysis been performed appropriately and rigorously?

Reviewer #1: Yes

Reviewer #2: Yes

3. Have the authors made all data underlying the findings in their manuscript fully available?

Reviewer #1: Yes

Reviewer #2: Yes

4. Is the manuscript presented in an intelligible fashion and written in standard English?

Reviewer #1: Yes

Reviewer #2: Yes

Reviewer #1: Comments to Authors

Technical

1. Explain how the integration of machine learning regression techniques contributes to the optical property prediction in the proposed PCF-SPR biosensor and with the machine leaning approaches can you be precise in finalizing the final sensor parameters.

2. I suggest presents a detail about various AI based techniques that can be used in the sensor optimization. Follow the following article to presents details of various AI based techniques used in sensor optimization and cite the article. "Finite Element Method-Based Modeling of a Novel Square Photonic Crystal Fiber Surface Plasmon Resonance Sensor with a Au–TiO2 Interface and the Relevance of Artificial Intelligence Techniques in Sensor Optimization"

3. Discuss the significance of the broad RI detection range (1.31 to 1.42) in the context of biosensing applications, justify the RI range since any RI below 1.33 is considered superficial, I suggest add a sentence like claiming theoretical analysis.

4. How does the use of SHAP (SHapley Additive exPlanations) enhance the interpretability of the ML model used for sensor design optimization?

5. Analyze how the achieved performance metrics such as wavelength sensitivity and figure of merit reflect the sensor's capability for high-precision detection.

6. Are these any sensing parameters which cannot be predicted using ML models, and why, and how do these predictions benefit the overall sensor design process?

7. Critically evaluate the influence of design parameters like Au thickness and pitch on the sensor's performance, as revealed by the SHAP analysis.

8. Compare the advantages of the hybrid design approach combining conventional methods and ML/XAI with traditional sensor optimization strategies in terms of efficiency and computational cost. Some prominent sensor models having merger of IMD and EMD, merger of Quasi D shaped with EMD approach etc need to be included in the comparison section of the article. Add a short paragraph suggesting future application of the sensor for various applications. Following are suggested references including these features.

https://doi.org/10.1007/s11468-024-02400-7

https://doi.org/10.1007/s11082-023-05016-z

https://doi.org/10.1016/j.ijleo.2022.169892

https://doi.org/10.1016/j.measurement.2021.110513

9. Justify the suitability of the proposed biosensor for medical diagnostics and chemical sensing, particularly in the context of cancer cell detection, refer the following reference.

https://doi.org/10.3390/bios15050292

Reviewer #2: Review for Authors

1. Explain how machine learning and artificial intelligence-based techniques can be used in the sensor design optimization and design. I suggest go through the following article and refer to various techniques used in the sensor modelling cite them too.

https://doi.org/10.3390/photonics12060565

https://doi.org/10.3390/bios15050292

2. The SHAP (SHapley Additive exPlanations) technique do not enhance the sensor detection capability, it provides more results to conclude, these models are used to analyze the sensor for design optimization purposes only.

3. In Figure 3 and section 2.2 can you see the in the text you have used D1, D2, D3 but in the diagram d1, d2, and where is d3, thus correct the notation. These create confusion.

4. Provide reference to Eq 1, 2, 4, 7, 9 etc, without proper reference it seems like you have developed the equation.

5. In Fig 12,13, 14 can you increase the font size of the x- and y- labels of the figure.

6. Fig 16 don’t look correct, format it well, format all image well increase the font size of the x-label and y-label for all figures.

7. Mention the applications of the proposed biosensor in real time from medical to environment application.

**Do you want your identity to be public for this peer review?** For information about this choice, including consent withdrawal, please see our Privacy Policy

Reviewer #1: No

Reviewer #2: **Yes: ** Dr. Ayushman Ramola, Ariel University, Israel

---

## [Author Response · Author response to Decision Letter 1]

5 Aug 2025

Responses to reviewers:

The authors would like to thank the reviewers for their precious time and valuable suggestions. We have carefully addressed and updated all the changes in the original manuscript. The following section provides point-by-point responses corresponding to all review comments.

#Reviewer 1

1. Explain how the integration of machine learning regression techniques contributes to the optical property prediction in the proposed PCF-SPR biosensor and with the machine leaning approaches can you be precise in finalizing the final sensor parameters.

Response: We have revised the manuscript to provide a more detailed explanation of how machine learning (ML) regression techniques contribute to the prediction of optical properties in the proposed PCF-SPR biosensor. A new paragraph has been added in the Section 2.6 (ML Algorithms).

2. I suggest presents a detail about various AI based techniques that can be used in the sensor optimization. Follow the following article to presents details of various AI based techniques used in sensor optimization and cite the article. "Finite Element Method-Based Modeling of a Novel Square Photonic Crystal Fiber Surface Plasmon Resonance Sensor with a Au–TiO2 Interface and the Relevance of Artificial Intelligence Techniques in Sensor Optimization"

Response:

In response, we have added a detailed discussion on various AI-based techniques used in sensor optimization, including Artificial Neural Networks (ANNs), classical machine learning models, ensemble methods, and deep learning approaches. Their applications in real-time prediction, inverse design, and performance enhancement have been highlighted. The recommended article has been thoroughly reviewed and cited appropriately in the revised manuscript see Section 2.5 (AI Techniques for PCF-SPR Sensor Optimization).

3. Discuss the significance of the broad RI detection range (1.31 to 1.42) in the context of biosensing applications, justify the RI range since any RI below 1.33 is considered superficial, I suggest add a sentence like claiming theoretical analysis.

Response: We have explained the reason for choosing the RI range (1.31–1.42) in Section 2.1. The lower part of the range (below 1.33) is included to cover some theoretical cases and very light or low-density samples.

4. How does the use of SHAP (Shapley Additive exPlanations) enhance the interpretability of the ML model used for sensor design optimization?

Response: We thank the reviewer for emphasizing the importance of model interpretability. To address this, we have added a clear explanation in Section 2.8 on how SHAP enhances the transparency of the ML model by identifying the impact of each input feature on the predicted optical properties. This helps to guide sensor design decisions and improves overall optimization.

5. Analyze how the achieved performance metrics such as wavelength sensitivity and figure of merit reflect the sensor's capability for high-precision detection.

Response: We have now included a detailed explanation in the revised manuscript in Section 4 (Discussion) to highlight how the achieved wavelength sensitivity and figure of merit (FOM) substantiate the high-precision detection capability of the proposed PCF-SPR biosensor. The newly added text elaborates on their implications in practical biosensing applications.

6. Are these any sensing parameters which cannot be predicted using ML models, and why, and how do these predictions benefit the overall sensor design process?

Response: We have addressed this point in Section 4 (Discussion) by discussing which sensing parameters are well-suited for ML prediction and which still require traditional simulation or experimental validation. This addition helps clarify the role of ML in the overall sensor design process and its practical limitations.

7. Critically evaluate the influence of design parameters like Au thickness and pitch on the sensor's performance, as revealed by the SHAP analysis.

Response: We have included a focused discussion in Section 4 (Discussion), based on SHAP analysis that critically evaluates the influence of design parameters such as gold thickness and pitch. Their respective impacts on key performance metrics like wavelength sensitivity and confinement loss are now clearly highlighted in the revised manuscript.

8. Compare the advantages of the hybrid design approach combining conventional methods and ML/XAI with traditional sensor optimization strategies in terms of efficiency and computational cost. Some prominent sensor models having merger of IMD and EMD, merger of Quasi D shaped with EMD approach etc need to be included in the comparison section of the article. Add a short paragraph suggesting future application of the sensor for various applications. Following are suggested references including these features.

https://doi.org/10.1007/s11468-024-02400-7

https://doi.org/10.1007/s11082-023-05016-z

https://doi.org/10.1016/j.ijleo.2022.169892

https://doi.org/10.1016/j.measurement.2021.110513

Response: Thank you for the valuable feedback. In response, we have expanded the comparison in Section 3.3 (Comparison with Previous Works) of the manuscript to highlight the advantages of our hybrid design strategy combining FEM-based simulation, machine learning (ML), and explainable AI (XAI) techniques over conventional sensor optimization methods. Additionally, we have included comparisons with prominent sensor architectures integrating IMD-EMD and Quasi D-shaped designs, as referenced. A paragraph outlining future real-time applications has also been added in Section 4 (Discussion).

9. Justify the suitability of the proposed biosensor for medical diagnostics and chemical sensing, particularly in the context of cancer cell detection, refer the following reference. https://doi.org/10.3390/bios15050292

Response: We have added this in Section 4 (Discussion), to further elaborate on the medical and chemical sensing relevance of the proposed PCF-SPR biosensor, with a focus on its applicability in cancer diagnostics. The suggested reference has been reviewed and appropriately cited to support our explanation.

#Reviewer 2

1. Explain how machine learning and artificial intelligence-based techniques can be used in the sensor design optimization and design. I suggest go through the following article and refer to various techniques used in the sensor modelling cite them too.

https://doi.org/10.3390/photonics12060565

https://doi.org/10.3390/bios15050292

Response: Thank you for the valuable suggestion. We have revised the manuscript to include a detailed overview of how ML and AI techniques contribute to PCF-SPR sensor optimization. This has been added under Section 2.5 (AI Techniques for PCF-SPR Sensor Optimization), along with relevant insights and citations from the suggested references.

2. The SHAP (SHapley Additive exPlanations) technique do not enhance the sensor detection capability, it provides more results to conclude, these models are used to analyze the sensor for design optimization purposes only.

Response: We agree with the reviewer’s observation. The manuscript has been updated in Section 2.8 (Explainable AI (XAI)) to clarify that SHAP does not directly improve sensing performance but helps interpret ML outputs, identify key design parameters, and support informed sensor optimization decisions.

3. In Figure 3 and section 2.2 can you see the in the text you have used D1, D2, D3 but in the diagram d1, d2, and where is d3, thus correct the notation. These create confusion.

Response: Thank you for pointing out the inconsistency. The notations in Figure 3 and Section 2.2 have been corrected for consistency. All diameter labels now uniformly use the same format (D1, D2, D3), and the missing label (D3) has been added to the figure to avoid confusion.

4. Provide reference to Eq 1, 2, 4, 7, 9 etc, without proper reference it seems like you have developed the equation.

Response: References for Equations 1, 2, 4, 7, 9, and others have been added in the revised manuscript to clearly indicate their original sources and avoid any confusion regarding authorship.

5. In Fig 12, 13, 14 can you increase the font size of the x- and y- labels of the figure.

Response: Thank you for the suggestion. The font size of the x- and y-axis labels in Figures 12, 13, and 14 has been increased to improve readability.

6. Fig 16 don’t look correct, format it well, format all image well increase the font size of the x-label and y-label for all figures.

Response: Thank you for pointing this out. Figure 16 has been reformatted for better clarity, and the font sizes of the x- and y-axis labels have been increased. Additionally, all other figures in the manuscript have been reviewed and formatted accordingly to ensure visual consistency and improved readability.

7. Mention the applications of the proposed biosensor in real time from medical to environment application.

Response: Thank you for your valuable suggestion. Real-time applications in medical diagnostics and environmental monitoring have been added in Section 4 (Discussion) to emphasize the sensor’s practical relevance.

---

## [Editor Report · Decision Letter 1]

8 Aug 2025

Design optimization of high-sensitivity PCF-SPR biosensor using machine learning and explainable AI

PONE-D-25-28444R1

Dear Dr. Khatun,

We’re pleased to inform you that your manuscript has been judged scientifically suitable for publication and will be formally accepted for publication once it meets all outstanding technical requirements.

Kind regards,

Yuan-Fong Chou Chau

Academic Editor

PLOS ONE
---

## [Editor Report · Acceptance letter]

PONE-D-25-28444R1

PLOS ONE

Dear Dr. Khatun,

I'm pleased to inform you that your manuscript has been deemed suitable for publication in PLOS ONE. Congratulations! Your manuscript is now being handed over to our production team.

Kind regards,

on behalf of

Dr. Yuan-Fong Chou Chau

Academic Editor

PLOS ONE